# BATCH PRUNING BY ACTIVATION STABILITY

**Md Mustakin Alam**[1][*]   **Shaker Islam**[2]   **Aminul Islam**[1]
[1]University of Louisiana at Lafayette    [2]BRAC University
{md-mustakin.alam1, aminul}@louisiana.edu, shaker.islam@g.bracu.ac.bd

## ABSTRACT

Training deep neural networks remains costly in terms of data, time, and energy, limiting their deployment in large-scale and resource-constrained settings. To address this, we propose Batch Pruning by Activation Stability (*B-PAS*), a dynamic plug-in strategy that accelerates training by removing batches that contribute less to learning. *B-PAS* monitors the stability of activation representations across epochs and prunes batches whose activation variance exhibits minimal change, indicating diminishing learning utility. Applied to ResNet-18, ResNet-50, and the Convolutional vision Transformer (CvT) on CIFAR-10, CIFAR-100, SVHN, and ImageNet-1K, *B-PAS* reduces training batch usage by up to 57% with no loss in accuracy, and by 47% while slightly improving accuracy. Moreover, it achieves up to 61% savings in GPU node-hours, outperforming prior state-of-the-art pruning methods with up to 29% higher data savings and 21% greater GPU node-hour savings. We further demonstrate the generalization of *B-PAS* by extending it to GPT-2 fine-tuning, showing that activation stability can serve as an effective pruning signal beyond vision models. These results highlight activation stability as a powerful internal signal for efficient training, offering a practical and sustainable path toward data and energy-efficient deep learning. Our code is publicly available at https://github.com/mustakinalam/Batch-Pruning-by-Activation-Stability.

## 1   INTRODUCTION

Deep learning has emerged as a powerful paradigm for solving complex tasks across a variety of domains. These models, while highly effective, are inherently resource and time-intensive, frequently consuming significant GPU hours and memory bandwidth during both training and inference phases. They utilize large amounts of computation even on redundant or less informative data, leading to inefficiencies in resource-constrained environments. Among deep learning models, Convolutional Neural Networks (CNNs) have achieved remarkable success in a wide range of computer vision tasks, including image classification, object detection, and segmentation (Krizhevsky et al., 2012; Simonyan & Zisserman, 2014; He et al., 2016; 2017; Law & Deng, 2018). However, these performance gains often come at the cost of increased computational demands. State-of-the-art CNN architectures typically require large-scale datasets such as ImageNet (Krizhevsky et al., 2012) for effective training and involve millions of parameters, making them expensive to train and deploy. This poses a significant barrier for practitioners with limited computational resources. Reducing training costs without sacrificing performance remains a longstanding challenge in deep learning.

A straightforward approach to address this challenge is to reduce the amount of training data. Techniques such as dataset distillation (Nguyen et al., 2021; Zhao & Bilen, 2023; Wang et al., 2022) and coreset selection (Har-Peled & Mazumdar, 2004; Park et al., 2022; Xia et al., 2022) aim to synthesize or select a compact, informative subset of the original dataset. While effective in reducing data volume, these methods often introduce nontrivial computational overhead and may result in degraded model performance. Weighted sampling methods (Zhao & Zhang, 2015; Csiba & Richtárik, 2018; Johnson & Guestrin, 2018) improve convergence by increasing the sampling frequency of informative samples, but their performance is highly sensitive to the choice of model and dataset.

Another line of research focuses on reducing the number of training iterations through data pruning. Static pruning methods estimate sample utility scores and remove low-utility samples before training

---

[*]Corresponding author

begins (Toneva et al., 2018; Paul et al., 2021), but often incur high preprocessing costs and lack adaptability during training. Dynamic pruning approaches mitigate these issues by adjusting the pruning process on the fly. For example, InfoBatch (Qin et al., 2024) dynamically prunes low-utility samples using a soft-pruning strategy combined with expectation rescaling to maintain unbiased gradients. Similarly, He et al. (2024) leverages prediction uncertainty and training dynamics to prune up to 25% of data from large-scale datasets such as ImageNet without sacrificing accuracy. However, these approaches often rely on complex heuristics.

Early stopping is a widely used strategy for reducing data usage by terminating training once performance plateaus (Duvenaud et al., 2016; Mahsereci et al., 2017; Bonet et al., 2021). However, most existing approaches rely on gradient-based signals that often fail to generalize across modern optimizers or require carefully tuned hyperparameters and specialized frameworks. More recently, Ahmad et al. (2024) proposed an early stopping criterion based on the stability of convolutional activations, highlighting the potential of internal network dynamics as reliable indicators of training progress. This line of work builds on the broader concept of activation stability, which is closely linked to the phenomenon of Neural Collapse (Papyan et al., 2020), wherein class-specific representations become increasingly aligned and activation patterns stabilize as training converges. Together, these findings suggest that activation stability offers a promising direction for analyzing data utility through internal model dynamics.

While prior data pruning approaches such as InfoBatch (Qin et al., 2024) rely on per-sample loss statistics, gradient rescaling, or fixed heuristics, often incurring additional computation or requiring explicit loss tracking, our work introduces a different perspective: *Can the stability of internal activations serve as a signal for assessing the informativeness of batches during training, thereby enabling effective and dynamic pruning of redundant batches?* We answer this affirmatively by introducing a lightweight, plug-and-play method that dynamically prunes low-utility batches based on the stability of activation variances across network layers. Unlike strategies that depend on difficulty scores, auxiliary models, or manually crafted rules, our approach is non-intrusive, as it operates on activation statistics already available from the forward pass, thereby introducing negligible overhead. The pruning is performed on-the-fly during training, without pretraining phases, validation labels, or static schedules. Our framework continuously monitors the mean standard deviation of flattened activations across layers for each batch across consecutive epochs and prunes batches if the change in this statistic is negligible across any neural network architecture. For the demonstration, applied to convolutional neural networks, this activation-driven pruning framework reduces redundant data usage across epochs while preserving training effectiveness and generality, offering a practical and efficient solution that establishes a foundation for extending stability-based pruning to broader deep learning architectures. Our key contributions are as follows:

**Activation Stability-Guided Dynamic Batch Pruning.** We introduce a lightweight, model-internal pruning strategy, 'Batch Pruning by Activation Stability' (*B-PAS*) that dynamically identifies and removes low-utility batches during training by leveraging the stability of activation representations. Specifically, we track the mean standard deviation of activations across hidden layers and consecutive epochs to assess whether a batch continues to contribute meaningful learning gradients. Batches exhibiting negligible change in activation variance are deemed redundant and pruned on-the-fly, without relying on auxiliary networks or handcrafted difficulty metrics. The pruning behavior is controlled by a tunable threshold hyperparameter $\delta$, enabling adaptability across datasets and model architectures.

**Comprehensive Evaluation on Benchmark Models.** We evaluate *B-PAS* on ResNet-18, ResNet-50, and CvT (Wu et al., 2021) across CIFAR-10, CIFAR-100 (Krizhevsky et al., a;b), SVHN (Netzer et al., 2011), and ImageNet-1K (Krizhevsky et al., 2012), with extensive threshold sweeps (45 $\delta$ settings on CIFAR-10 and eight on ImageNet-1K). Results show that *B-PAS* prunes up to 57% of training batches without accuracy loss, and up to 47% while slightly improving accuracy, while reducing computational cost by as much as 61% in GPU node-hours. These findings highlight both the robustness and tunability of activation stability as a pruning signal, delivering substantial training efficiency gains across scales and architectures. We further extend *B-PAS* to GPT-2 fine-tuning on the Alpaca dataset, demonstrating its applicability to LLMs and text-based tasks.

**Data Savings Index (DSI).** We introduce DSI, a new metric that quantifies the cumulative fraction of training data saved during learning. DSI provides a direct measure of data efficiency, with higher values indicating greater reductions in training cost and computational resources.

## 2   BATCH PRUNING BY ACTIVATION STABILITY (*B-PAS*)

In this section, we introduce the concept of a plug-in to CNN architectures: Batch Pruning by Activation Stability (*B-PAS*), detailing its formulation and adaptation for image applications. Figure 1 illustrates the working mechanism of a conventional CNN (labeled 'A') alongside the proposed plug-in (labeled 'B').

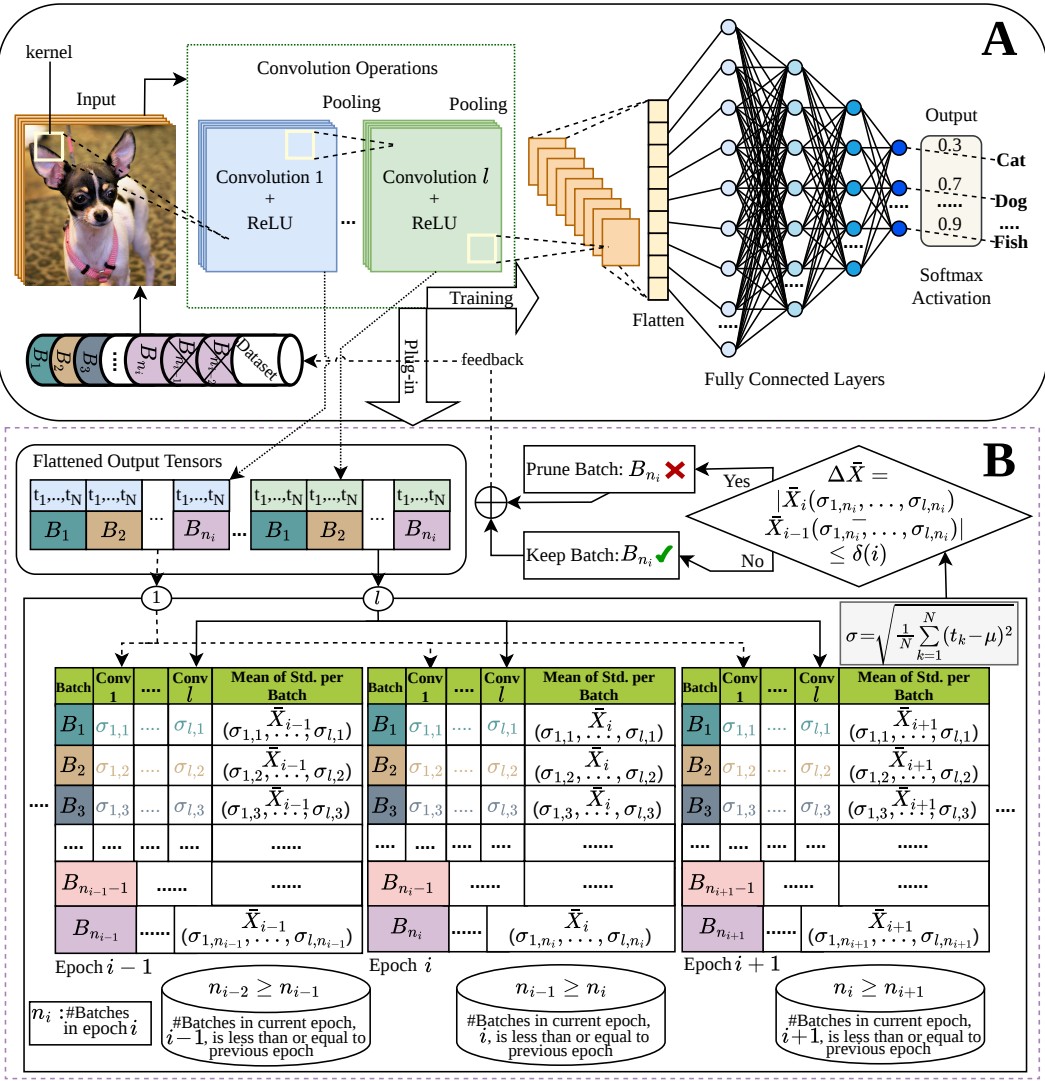

Figure 1: Overview of the proposed Batch Pruning by Activation Stability *(B-PAS)* plug-in integrated into a prevalent Convolutional Neural Network (CNN) training pipeline. (A) The conventional CNN architecture processes input images in batches through convolutional, ReLU activation, and pooling layers, followed by a fully connected classifier. (B) The *B-PAS* module monitors the standard deviation of ReLU-activated convolutional outputs for each batch across epochs. For each epoch, the standard deviation is recorded per convolutional layer (columns) and aggregated across layers to compute the mean standard deviation for each batch (rows). These per-batch means are then compared across consecutive epochs to assess activation stability. If the change ($\Delta \bar{X}$) for a batch $B_{n_i}$ falls below a dynamic threshold $\delta(i)$, the batch is deemed to have low learning utility and is pruned from subsequent training. By updating the dataset at the end of each epoch using this feedback, the process dynamically focuses training on batches that continue to provide meaningful gradient information, thereby improving efficiency without compromising accuracy.

## 2.1 ACTIVATION STABILITY

Building on the observation from Neural Collapse (Papyan et al., 2020) that activation patterns stabilize as training converges, we adopt the idea of (Ahmad et al., 2024), which links data variation across CNN layers to near-optimal learning capacity. Extending this principle to the batch level, we observe that as training progresses, certain batches show diminishing changes in activation variance across consecutive epochs, signaling little additional learning. To capture this effect, we use the widely adopted Rectified Linear Unit (ReLU) (Agarap, 2018), where the variance of ReLU-activated outputs provides a meaningful measure of feature stability. When this variance remains nearly unchanged across epochs, the corresponding batch is deemed converged, as its contribution to weight updates has effectively stabilized.

We compute variance after ReLU since it not only introduces non-linearity but also suppresses inactive neurons, ensuring that variance reflects sparse, meaningful features rather than noisy pre-activation values. Thus, monitoring the standard deviation of post-ReLU activations provides a reliable estimate of batch learning utility: once changes become negligible, the batch is pruned. This enables the model to focus computation on informative batches while discarding redundant ones, improving efficiency without harming performance.

In Figure 1(A), the 'Convolution Operations' module illustrates the ReLU-activated convolutional feature maps of input images, organized batch-wise across training epochs. At each convolutional layer, training images are processed in batches denoted by $B_1, \ldots, B_{n_i}$, where $B_{n_i}$ refers to the final batch in epoch $i$. For instance, if epoch $i$ contains 400 batches, then $B_{n_i} = B_{400}$, indicating that the $400^{\text{th}}$ batch is the last batch of that epoch. Each batch consists of multiple images, which are represented as tensors containing numerical activation values. To quantify activation variability, we compute the standard deviation of these values by first flattening the output tensors of each ReLU-activated convolutional layer. The standard deviation is then computed using $\sigma = \sqrt{\frac{1}{N} \sum_{k=1}^{N} (t_k - \mu)^2}$, where $t_k$ denotes each individual value from the flattened tensor (for $k = 1, \ldots, N$), $\mu$ is the mean of these values, $N$ is the total number of values in the flattened tensor, and $\sigma$ represents the resulting standard deviation. This computation is performed independently for each batch and each convolutional layer.

In Figure 1(B), each epoch table (i.e., Epochs $i-1$, $i$, and $i+1$) presents the standard deviation of the ReLU-activated outputs from the convolutional layers. In each table, rows correspond to data batches (e.g., $B_1, \ldots, B_{n_i}$), and columns represent individual convolutional layers, except the final column. The last column contains the mean standard deviation for each batch, computed across all convolutional layers. For example, the row corresponding to batch $B_{n_i}$ includes standard deviations $\sigma_{1,n_i}, \ldots, \sigma_{l,n_i}$, where $\sigma_{l,n_i}$ denotes the standard deviation of the output from the $l$-th convolutional layer for batch $B_{n_i}$, and $l$ is the total number of convolutional layers. The mean standard deviation for the final batch $B_{n_i}$ in epoch $i$ is denoted as $\bar{X}_i(\sigma_{1,n_i}, \ldots, \sigma_{l,n_i})$, indicating the mean standard deviation across all $l$ convolutional layers.

These mean standard deviations are used to track the variance behavior of each batch over time. Beginning with epoch one, the mean standard deviation is computed for every batch. From epoch two onward, the current epoch's mean is compared against the previous epoch's mean for each batch. If the change is negligible, the batch is considered to have converged and may be pruned from subsequent training epochs. This pruning process continues iteratively for the remaining epochs.

## 2.2 BATCH PRUNING

The decision to prune a batch is based on the stability of its mean activation standard deviation across consecutive epochs. As illustrated in Figure 1(B), we consider three epochs: $i-1$, $i$, and $i+1$. If the absolute difference between these means across consecutive epochs falls below a dynamic threshold $\delta(i)$, the batch is deemed to have converged and is excluded from subsequent training.

To illustrate, consider the standard deviations of batch $B_{n_i}$ across the $l$ convolutional layers as $\sigma_{1,n_i}, \ldots, \sigma_{l,n_i}$. The mean standard deviation for this batch at epoch $i$ is denoted by $\bar{X}_i(\sigma_{1,n_i}, \ldots, \sigma_{l,n_i})$, and at epoch $i-1$ by $\bar{X}_{i-1}(\sigma_{1,n_i}, \ldots, \sigma_{l,n_i})$. The change in mean standard deviation/variance for the batch $B_{n_i}$ is computed as:

$$\Delta \bar{X} = \left| \bar{X}_i(\sigma_{1,n_i}, \ldots, \sigma_{l,n_i}) - \bar{X}_{i-1}(\sigma_{1,n_i}, \ldots, \sigma_{l,n_i}) \right| \leq \delta(i)$$

If $\Delta\bar{X}$ for the batch $B_{n_i}$ falls below the threshold $\delta(i)$, the batch is pruned and excluded from training in epoch $i+1$. This criterion is applied to all batches at the end of each epoch. The pruning decisions update the dataset by retaining only the informative batches, resulting in a reduced set of training data for the next epoch and a progressively more efficient training process. As training progresses, the number of retained batches decreases monotonically. Formally, for any epoch $i$, the number of batches $n_i$ satisfies $n_i \leq n_{i-1}$. Because the pruning is dynamic, the composition of the final batch may change across epochs. For instance, for epoch $i$, let the last batch be $B_{n_i}$ and the second last $B_{n_i-1}$; similarly, in epoch $i-1$, they are $B_{n_{i-1}}$ and $B_{n_{i-1}-1}$. Without pruning, these batches are identical, whereas pruning one batch in epoch $i-1$ shifts the indexing so that in epoch $i$, $B_{n_i} = B_{n_{i-1}-1}$.

## 2.3 DYNAMIC THRESHOLD FOR PRUNING

To determine the negligible difference between activation variances across epochs, we introduce a tolerance hyperparameter $\delta$ to formalize pruning. Instead of requiring exact invariance, $\delta$ captures variance saturation with a small non-zero margin.

We empirically analyze the effect of different $\delta$ values on pruning dynamics. Very small $\delta$ leads to overly conservative pruning, discarding batches only when variance changes are nearly imperceptible, thus retaining most batches. Conversely, large $\delta$ induces aggressive pruning, prematurely removing informative batches and risking early training collapse.

To balance this trade-off, we adopt a dynamic schedule: $\delta(i) = \delta_s \cdot e^{\alpha i}$, $\alpha = \frac{1}{I}\ln\left(\frac{\delta_e}{\delta_s}\right)$, where $\delta_s$ and $\delta_e$ are the initial and final threshold values, $i$ is the current epoch, and $I$ is the total number of epochs. This schedule is conservative in early train-

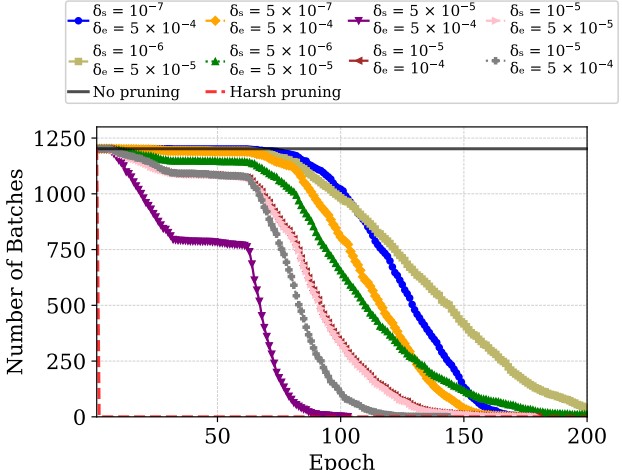

Figure 2: Pruning dynamics under different $\delta$ settings for ResNet-50 on ImageNet-1K (200 epochs). Lower thresholds (e.g., $\delta \in [10^{-6}, 5\times10^{-5}]$) lead to conservative pruning, retaining most batches until late epochs, while higher thresholds (e.g., $\delta \in [5\times10^{-5}, 5\times10^{-4}]$) cause aggressive pruning and premature training termination. The dynamic schedule ($\delta_s = 5\times10^{-6}, \delta_e = 5\times10^{-5}$) provides a balanced trajectory, steadily reducing data.

ing, when features are broadly learned, and more aggressive in later stages, when learning stabilizes. At epoch $i$, batches with $\Delta\bar{X} < \delta(i)$ are pruned.

Figure 2 illustrates these dynamics using eight sets of $\delta$ values on ImageNet-1K with ResNet-50. The horizontal line shows no pruning, while the sharp drop at epoch 2 depicts extreme aggressiveness (all batches removed). Lower ranges (e.g., $\delta \in [10^{-6}, 5\times10^{-5}]$) yield delayed, conservative pruning, whereas higher ranges (e.g., $\delta \in [5\times10^{-5}, 5\times10^{-4}]$) prune aggressively and often terminate training early. A balanced configuration of $\delta_s = 5\times10^{-6}$ and $\delta_e = 5\times10^{-5}$ achieves steady pruning without sacrificing stability. Nonetheless, $\delta$ remains a tunable hyperparameter, adaptable to dataset complexity, architecture, and resource budgets.

## 3 EXPERIMENT

We evaluate *B-PAS* as a plug-in module on ResNet-18, ResNet-50, and the Convolutional vision Transformer (CvT) to assess robustness and generality. In the networks, we add a Batch Normalization layer, which normalizes per-batch variance and has a strong impact on pruning dynamics. Experiments span CIFAR-10, CIFAR-100, SVHN ($32\times32$), and ImageNet-1K ($\sim$1.3M images, $224\times224$). Unless specified, we use $\delta \in [10^{-6}, 5 \times 10^{-5}]$ for CIFAR-10/100 and SVHN, and $\delta \in [5 \times 10^{-6}, 5 \times 10^{-5}]$ for ImageNet-1K, corresponding to empirically validated balanced pruning regimes. To further demonstrate the generalization of *B-PAS*, it is extended to GPT-2 fine-tuning with the Alpaca dataset. For evaluation, we report validation accuracy and GPU node-hours, but since hardware and system

factors confound GPU time, we also introduce the Data Savings Index (DSI), which quantifies skipped data across epochs and batches as a continuous measure of efficiency. To ensure well-defined variance tracking, batches are fixed once at initialization (rather than regenerated each epoch) while intra-batch shuffling is applied to preserve stochasticity and avoid overfitting. In contrast, full-dataset baseline training follows the standard batch shuffling protocol. Full experimental details are provided in Appendix A.1.

**GPU node-hours.** It is a measure of computational cost, defined as GPU node-hours $= g \times h$ where $g$ is the number of GPUs used during training and $h$ is the total training time (in hours). The percentage of node-hours saved is reported relative to the full dataset baseline.

**Data Savings Index (DSI).** To directly quantify the reduction in training data usage, we introduce the Data Savings Index: $\text{DSI} = 1 - \frac{\sum_{i=1}^{e_s} n_i}{e_0 \cdot n_0}$, where $n_i$ is the number of batches retained in epoch $i$, $n_0$ is the total number of batches before training begins, $e_s$ is the epoch at which training stops, and $e_0$ is the scheduled number of epochs in the absence of pruning or early stopping ($e_s \leq e_0$). The DSI value lies in $[0, 1]$, with higher values indicating greater savings in data usage. For example, if a model is scheduled to train for 5 epochs with 200 batches per epoch, but training stops at epoch 3 after processing 200, 190, and 180 batches, the DSI is $1 - \frac{200+190+180}{5 \times 200} = 0.43$, meaning that $43\%$ of the potential training data is saved.

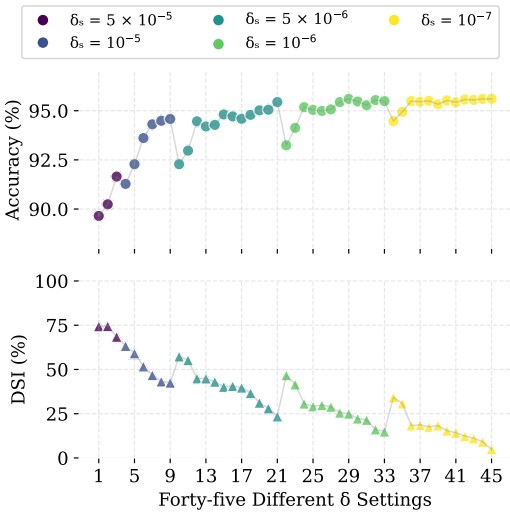

Figure 3: Empirical analysis of *B-PAS* on CIFAR-10 with ResNet-18 across 45 $\delta$ settings. Smaller thresholds have less data savings with higher accuracy, larger thresholds save more data at the cost of accuracy, and intermediate values (e.g., $\delta \in [10^{-6}, 5 \times 10^{-5}]$) provide the best trade-off.

## 3.1 RESULT ANALYSIS

**Empirical Analysis on CIFAR-10.**

We first evaluate *B-PAS* using ResNet-18 on CIFAR-10 across forty-five threshold ($\delta$) settings. As shown in Figure 3, we define five groups of starting thresholds ($\delta_s$), each paired with multiple end thresholds ($\delta_e$). Specifically, for $\delta_s = 5 \times 10^{-5}$ and $\delta_s = 10^{-5}$, we consider three and six values of $\delta_e$, respectively, while for the remaining groups we test 12 values. The $\delta_e$ values are $\{10^{-3}, 5 \times 10^{-4}, 10^{-4}, 9 \times 10^{-5}, 8 \times 10^{-5}, 7 \times 10^{-5}, 6 \times 10^{-5}, 5 \times 10^{-5}, 4 \times 10^{-5}, 3 \times 10^{-5}, 2 \times 10^{-5}, 10^{-5}\}$. For the $\delta_s = 5 \times 10^{-5}$ group, only $\delta_e = \{10^{-3}, 5 \times 10^{-4}, 10^{-4}\}$ are included; for $\delta_s = 10^{-5}$, we use $\delta_e = \{10^{-3}, 5 \times 10^{-4}, 10^{-4}, 5 \times 10^{-5}, 4 \times 10^{-5}, 3 \times 10^{-5}\}$. The results reveal a clear pattern in accuracy and DSI percentage. Larger threshold groups (e.g., $\delta_s = 5 \times 10^{-5}$) yield very high data savings but also sacrifice accuracy. Conversely, smaller threshold groups (e.g., $\delta_s = 10^{-7}$) retain nearly all data, resulting in

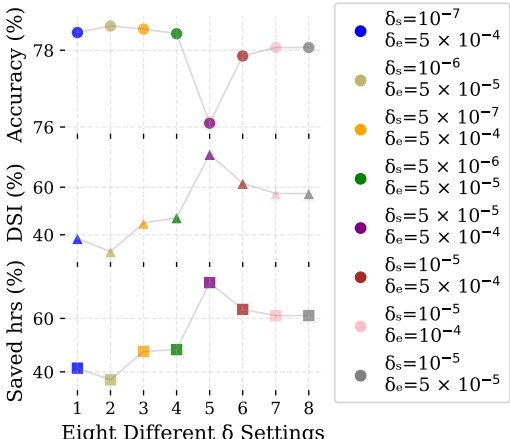

Figure 4: Empirical analysis of *B-PAS* on ImageNet-1K with ResNet-50 across eight $\delta$ settings. Smaller thresholds retain most data with higher accuracy but lower savings, larger thresholds prune aggressively and harm accuracy, while intermediate values (e.g., $\delta \in [5 \times 10^{-6}, 5 \times 10^{-5}]$) achieve the best trade-off.

high accuracy but reduced pruning benefits. Values in intermediate groups (e.g., $\delta_s = 10^{-6}$) provide a balanced trade-off. In particular, the 29th setting with $\delta \in [10^{-6}, 5 \times 10^{-5}]$ achieves the highest accuracy while saving data by $25\%$. Thus, for CIFAR-10 and similar $32 \times 32$ datasets, $\delta_s = 10^{-6}$ and $\delta_e = 5 \times 10^{-5}$ represent a strong default setting, as further validated in Table 3.

**Empirical Analysis on ImageNet-1K.** Guided by the CIFAR-10 analysis, we evaluate *B-PAS* with ResNet-50 on ImageNet-1K over 200 epochs across eight threshold settings. As illustrated in Figure 4, we observe the same trend: smaller thresholds preserve accuracy but maintain low DSI and GPU node-hours savings, while larger thresholds lead to aggressive pruning and accuracy degradation. For example, $\delta \in [10^{-6}, 5 \times 10^{-5}]$ yields the highest accuracy but also the lowest data and GPU node-hours savings. In contrast, $\delta \in [5 \times 10^{-5}, 5 \times 10^{-4}]$ leads to early, aggressive pruning, often terminating training around epoch 100, resulting in very low accuracy. This behavior aligns with the pruning dynamics shown in Figure 2: smaller thresholds delay pruning, retaining most batches until late epochs, while larger thresholds trigger sharp drops of batches earlier in training. The most balanced results are obtained for $\delta \in [5 \times 10^{-6}, 5 \times 10^{-5}]$, as supported by both Figure 2 and Figure 4.

Table 1: Comparison of *B-PAS* against InfoBatch and random pruning on CIFAR-10/100 using ResNet-18/50. *B-PAS* matches or exceeds InfoBatch across different pruning ratios while maintaining the accuracy of the full dataset, showing its competitiveness on small-scale datasets.

| Approach | CIFAR-10 | | CIFAR-100 | |
|---|---|---|---|---|
| in % $\Rightarrow$ | DSI | Acc | DSI | Acc |
| ResNet-18 | 0 | **95.60±0.2** | 0 | **78.20±0.3** |
| +Random | 25 | 94.60±0.3 | 24 | 75.36±0.4 |
| +InfoBatch (30%) | 22 | **95.60±0.1** | 19 | **78.20±0.1** |
| +InfoBatch (50%) | 37 | 95.10±0.3 | 32 | 78.10±0.1 |
| +InfoBatch (70%) | 53 | 94.70±0.4 | 47 | 76.50±0.4 |
| +*B-PAS* | 25 | **95.60±0.1** | 24 | **78.20±0.1** |
| ResNet-50 | 0 | **95.66±0.1** | 0 | **80.60±0.5** |
| +Random | 33 | 94.50±0.3 | 30 | 75.77±0.4 |
| +InfoBatch (30%) | 21 | **95.66±0.1** | 18 | **80.60±0.1** |
| +InfoBatch (50%) | 36 | 95.20±0.3 | 32 | 80.05±0.1 |
| +InfoBatch (70%) | 52 | 94.99±0.4 | 45 | 79.37±0.4 |
| +*B-PAS* | 33 | **95.66±0.1** | 30 | **80.60±0.5** |

This setting achieves near-maximum accuracy with DSI over 45% and saved GPU node-hours by 48%, indicating efficient pruning without compromising performance. Moreover, although the choice of the threshold hyperparameter $\delta$ may appear to require extensive tuning for new datasets or model families, we find that full training is unnecessary for calibration. In practice, reliable $\delta$ values can be obtained by training on only 10% of the data, enabling fast and efficient hyperparameter selection. Detailed analyses are provided in Appendix A.4. An additional observation is that DSI and GPU Node-hour savings follow the same trend across experiments, confirming that data usage is tightly coupled to training cost. Unlike node-hours, however, DSI provides a more comprehensive and system-independent measure by capturing saved data across epochs.

**Comparison with SOTA Method.**

We compare *B-PAS* with the state-of-the-art pruning approach Info-Batch (Qin et al., 2024) on CIFAR-10, CIFAR-100, and ImageNet-1K (Tables 1 and 2). InfoBatch has previously demonstrated superiority over 14 static and three dynamic pruning baselines, establishing it as a strong reference point (A Comprehensive tabular comparison is provided in Appendix A.5.). On CIFAR-10 and CIFAR-100 (Table 1), *B-PAS* achieves accuracy comparable to the full dataset and Info-Batch across multiple pruning ratios.

Table 2: Comparison of *B-PAS* and InfoBatch on ImageNet-1K with ResNet-50. *B-PAS* achieves greater efficiency, reducing data usage by 57% and node-hours by 61%, while preserving accuracy at the full dataset level. More conservative $\delta$ values yield further accuracy gains (78.43%) with moderate savings, highlighting the scalability of activation stability signals in large-scale training.

| Approach | ImageNet-1K | | |
|---|---|---|---|
| in % $\Rightarrow$ | Saved hrs | DSI | Acc |
| ResNet-50 | 0 | 0 | 78.07±0.1 |
| +InfoBatch (40%) | 40 | 28 | 78.07±0.1 |
| +*B-PAS* ($\delta \in [10^{-5}, 10^{-4}]$) | **61** | **57** | 78.07±0.1 |
| +*B-PAS* ($\delta \in [5 \times 10^{-6}, 5 \times 10^{-5}]$) | 48 | 47 | **78.43±0.1** |

For example, in CIFAR-100 with ResNet-50, *B-PAS* saves 30% of the data while maintaining 80.6% accuracy, closely matching InfoBatch at the 30% pruning ratio (DSI = 18%). To further assess pruning behavior, we introduce a random pruning baseline: we track the number of batches flagged by activation stability but prune the same number of batches chosen at random. Unlike *B-PAS*, random pruning consistently degrades accuracy, underscoring that activation stability identifies non-informative batches rather than simply reducing training data. These results demonstrate that *B-PAS* is competitive on small-scale datasets, with modest but reliable savings due to the limited learning utility in low-resolution tasks. In contrast, the advantage of *B-PAS* becomes more pronounced on ImageNet-1K. As shown in Table 2, InfoBatch achieves 28% data savings and 40% GPU node-hour reduction while maintaining full dataset baseline accuracy (78.07%). By comparison, *B-PAS* delivers substantially larger gains: with $\delta \in [10^{-5}, 10^{-4}]$, it saves 57% of the data and 61% of node-hours

at the same accuracy, and with more conservative thresholds ($\delta \in [5 \times 10^{-6}, 5 \times 10^{-5}]$), it further improves accuracy to $78.43\%$ while still achieving $47\%$ data savings and $48\%$ node-hour reduction. These results highlight that activation stability-driven pruning not only matches InfoBatch in accuracy preservation but also provides significantly greater efficiency on large-scale training.

This demonstrates that activation stability provides a scalable pruning signal in large-scale training, where the learning utility of data is more significant. Finally, just as InfoBatch evaluates multiple pruning ratios (30%, 50%, 70%), *B-PAS* naturally supports different pruning regimes through the choice of $\delta$ values. Smaller $\delta$ ranges yield conservative pruning with lower DSI, while larger values trigger more aggressive pruning and faster convergence. This flexibility allows *B-PAS* to adapt pruning aggressiveness without requiring handcrafted schedules or explicit loss tracking, underscoring its practical utility in both small- and large-scale settings.

**Cross-Architecture, Cross-Task and Dataset Robustness.** Table 3 evaluates the generalization of *B-PAS* across diverse architectures (ResNet-18, ResNet-50, and CvT) and datasets (CIFAR-10, CIFAR-100, SVHN, and ImageNet-1K). Several consistent trends emerge.

First, across CNN architectures, *B-PAS* preserves full dataset baseline accuracy while significantly reducing data usage and training cost. For example, on ImageNet-1K with ResNet-50, *B-PAS* achieves 78.43% accuracy (slightly higher than full dataset) while saving data usage by 47% and node-hours by 48%. Similar savings are observed on smaller datasets, with up to

Table 3: Cross-architecture and dataset robustness of *B-PAS* on CIFAR-10/100, SVHN, and ImageNet-1K with ResNet-18/50 and CvT. *B-PAS* preserves accuracy across models while reducing data usage and GPU node-hours.

| | CIFAR-10 | | CIFAR-100 | | SVHN | | ImageNet-1K | | |
|---|---|---|---|---|---|---|---|---|---|
| | R-18 | R-50 | R-18 | R-50 | R-18 | R-50 | R-18 | R-50 | CvT |
| Full Dataset | 95.60 | 95.66 | 78.2 | 80.6 | 95.90 | 96.27 | 70.05 | 78.07 | 79.65 |
| *B-PAS* | 95.60 | 95.66 | 78.2 | 80.6 | 95.97 | 96.27 | 71.5 | 78.43 | 79.6 |
| DSI(%) | 25 | 33 | 24 | 30 | 27 | 30 | 37 | 47 | 14 |
| Saved hrs(%) | 23 | 29 | 22 | 29 | 25 | 33 | 61 | 48 | 13 |

$33\%$ data usage savings on CIFAR-10 and $33\%$ node-hours reduction on SVHN. These results demonstrate that activation stability provides a reliable pruning signal across scales and architectures within the CNN family.

Extending beyond CNNs, we evaluate *B-PAS* on CvT-13. Pruning is performed by tracking stage-wise activations: after each CvT stage, token sequences are projected back to spatial format to compute variance on the multi-scale feature maps. With a conservative threshold range ($\delta \in [1 \times 10^{-5}, 5 \times 10^{-4}]$), pruning

Table 4: Comparison of full-data training and *B-PAS* on loss, perplexity, training time, and pruning statistics for GPT-2 large.

| | Loss | Perplexity | Avg. Epoch Time (s) | Total Time (s) | Pruned Batch (%) | DSI (%) |
|---|---|---|---|---|---|---|
| Full Data | 0.2207 | **1.25** | 5359.88 | 54420.13 | – | – |
| *B-PAS* | **0.2201** | **1.25** | **5039.11** | **51211.29** | 23.00 | 6 |

remains limited, achieving only $14\%$ DSI and $13\%$ GPU node-hour savings while preserving near-baseline accuracy. Using a more aggressive schedule ($\delta_s = 10^{-4}, \delta_e = 10^{-3}$), *B-PAS* attains substantially higher efficiency, reaching $35\%$ DSI and GPU-hour savings with only a modest accuracy drop ($\sim 79\%$). This behavior reflects the slower and noisier activation dynamics of transformer-based models, where stability emerges later than in CNNs. Since our CvT training spans 200 epochs, shorter than the 300+ epochs typically required for full convergence on ImageNet-1K, stabilization is delayed, naturally limiting pruning under milder thresholds. These results indicate that effective pruning with activation stability in CvT requires longer training or more aggressive $\delta$ schedules.

Table 4 reports the results of applying *B-PAS* during fine-tuning of GPT-2 large on the Alpaca instruction tuning dataset. Because this experiment involves a short 10-epoch fine-tuning run on a transformer-based LLM, we adopt a more aggressive pruning threshold range ($\delta_s = 10^{-3}, \delta_e = 10^{-2}$) to compensate for the smoother activation dynamics of transformers. As expected, activation stabilization occurs later in training, resulting in a lower Data Savings Index (DSI) compared to our large-scale vision experiments; nevertheless, *B-PAS* prunes 23% of batches. Crucially, pruning does not harm model performance: loss and perplexity remain unchanged or slightly improved. At the same

time, total training time is reduced by about an hour on $2\times$A100 GPUs. These findings demonstrate that *B-PAS* is compatible with transformer-based language models and provides measurable efficiency gains even in smaller fine-tuning settings. GPT-2 implementation details are in Appendix A.1.

## 3.2 ABLATION STUDIES

To better understand the behavior of *B-PAS*, we perform controlled ablation studies across learning rates, normalization layers, training epochs, pruning granularity, and optimizers. These studies highlight both the robustness of the method and the factors influencing its efficiency.

**Effect of Learning Rate.** Table 5 shows the impact of different learning rates on ImageNet-1K with ResNet-50. For a batch size of 256, we consider learning rates of 0.2, 0.01, and 0.1. While all settings achieve comparable DSI (45%–47%), accuracy varies significantly: 74.29% at LR $= 0.01$, 77.27% at LR $= 0.2$, and 78.43% at LR $= 0.1$. These results suggest that excessively small learning rates hinder convergence, while overly large ones reduce generalization. Importantly, the pruning dynamics of *B-PAS* remain stable across learning rates, showing that activation stability is largely agnostic to optimizer step size.

Table 5: *B-PAS* under different learning rates (LR).

| LR | DSI(%) | Acc(%) |
|----|--------|--------|
| 0.2 | 45 | 77.27 |
| 0.01 | 46 | 74.29 |
| 0.1 | 47 | 78.43 |

Table 6: Effect of Batch Normalization on *B-PAS*.

| | DSI(%) | Acc(%) |
|----|--------|--------|
| -BN | 19.72 | 89.87 |
| +BN | 25 | 95.60 |

Table 7: Effect of training epochs on *B-PAS* on ImageNet-1k.

| Epochs | DSI(%) | Acc(%) |
|--------|--------|--------|
| 90 | 12 | 78.07 |
| 200 | 47 | 78.43 |

Table 8: *B-PAS* pruning granularity.

| Pruning | Acc(%) |
|---------|--------|
| Sample | 70.87 |
| Batch | 78.43 |

**Impact of Batch Normalization.** Table 6 compares CIFAR-10 results with ResNet-18 with and without Batch Normalization (BN). Since BN normalizes feature statistics per batch, it strongly affects *B-PAS*. Without BN, activation trajectories are unstable, requiring more aggressive thresholds (e.g., $\delta_s = 5 \times 10^{-5}$, $\delta_e = 10^{-3}$) to obtain reasonable DSI (19.72%); with the default $\delta$ values, pruning is minimal (DSI $= 2\%$). Also, removing BN results in an accuracy drop. In contrast, with BN, the default thresholds yield effective pruning (DSI $= 25\%$) while improving accuracy to 95.60%. These results show that BN not only stabilizes activations but also enhances the discriminative signal used by *B-PAS*, enabling more reliable identification of redundant batches. More analyses are provided in Appendix A.7.

**Effect of Training Epochs.** Table 7 compares ImageNet-1K with ResNet-50 performance at 90 and 200 epochs. With 90 epochs, pruning remains limited (DSI $= 12\%$) due to insufficient stabilization, whereas at 200 epochs DSI rises to $47\%$ with improved accuracy ($78.43\%$). This highlights that longer training naturally allows greater pruning, where data savings become more critical and *B-PAS* proves most effective.

**Pruning Granularity: Batch vs. Sample Level.** Table 8 compares pruning at the sample and batch levels for ImageNet-1K on ResNet-50. For sample-level pruning, the accuracy reduces to 70.87%, compared to 78.43% for batch-level pruning. This discrepancy arises because sample-level pruning may disproportionately eliminate certain classes, leading to class imbalance and degraded generalization. In contrast, batch-level pruning preserves class diversity while still removing redundant information, confirming its superiority as the granularity choice for *B-PAS*.

Table 9: *B-PAS* preserves accuracy while achieving similar DSI across optimizers on CIFAR-10 with ResNet-18, demonstrating robustness to optimization dynamics.

| | SGD | Adam | AdaGrad |
|----|-----|------|---------|
| Full Dataset | 95.60 | 93.36 | 92.93 |
| *B-PAS* | 95.60 | 93.34 | 92.93 |
| DSI (%) | 25 | 24 | 22 |

**Optimizer Robustness.** Finally, Table 9 explores different optimizers on CIFAR-10 with ResNet-18. Across SGD, Adam, and AdaGrad (Ruder, 2016), *B-PAS* maintains accuracy identical to baseline training, with DSI ranging from 22% to 25%. These results highlight that the pruning criterion is robust to different optimization dynamics, even when another first-order adaptive optimizer (Adam/AdaGrad) alters activation trajectories. This reinforces that *B-PAS* generalizes across diverse optimization regimes without requiring re-tuning.

## 4 RELATED WORK

This work is related to three major research directions in efficient deep learning: static data pruning, dynamic data pruning, and dataset distillation. **Static pruning** methods select training subsets prior to training using criteria like geometric diversity (Sener & Savarese, 2017; Agarwal et al., 2020), uncertainty (Coleman et al., 2019; Gal et al., 2017), or learning difficulty (Toneva et al., 2018; Paul et al., 2021). Gradient-based approaches (e.g., GraNd, EL2N (Paul et al., 2021)) and decision-boundary methods like DeepFool (Ducoffe & Precioso, 2018) assess sample importance more directly. Other strategies include bilevel optimization (Killamsetty et al., 2021), submodular selection (Iyer et al., 2021), ensemble heuristics (Xia et al., 2022), and diversity-aware methods (Welling, 2009; Zheng et al., 2023). These techniques often require full dataset access and heavy pre-computation, limiting scalability. **Dynamic pruning** eliminates low-utility samples during training via online signals. Bandit-based methods (Raju et al., 2021), soft pruning with gradient rescaling (InfoBatch Qin et al. (2024)), and uncertainty-driven pruning (He et al., 2024) have shown success but often involve complex heuristics and tuning. **Dataset distillation** synthesizes compact training sets via gradient (Zhao & Bilen, 2023; Liu et al., 2023; Cui et al., 2023; Yang et al., 2023), distribution (Wang et al., 2022; Sajedi et al., 2023), or trajectory matching (Cazenavette et al., 2022; Du et al., 2023; Guo et al., 2024), yet faces challenges in scaling to high-resolution data and large models.

## 5 DISCUSSION AND CONCLUSION

Recent work, such as InfoBatch (Qin et al., 2024), advances data pruning through temporary, sample-level pruning with gradient rescaling to preserve unbiased gradient estimates. While effective, this approach requires maintaining per-sample loss statistics and revisiting all data in subsequent epochs, limiting efficiency in large-scale training. In contrast, our proposed *B-PAS* performs permanent, batch-level pruning based on activation stability, eliminating the need for gradient rescaling or repeated access to discarded data. This design significantly reduces data usage, training time, and GPU node-hours by leveraging internal activation dynamics instead of loss signals. A common con-

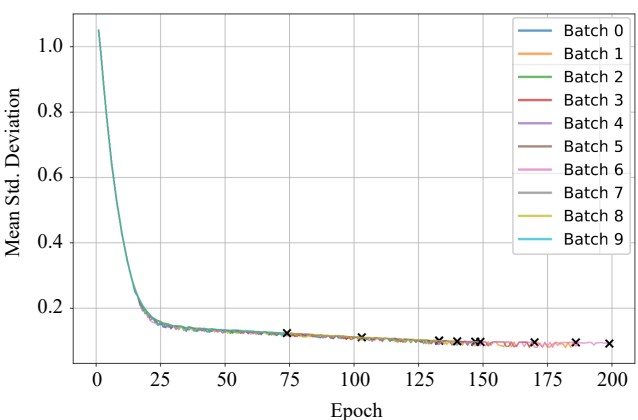

Figure 5: Declining mean standard deviation change ($\Delta \bar{X}$) across epochs for some selected batches on CIFAR-10 with ResNet-18, indicating diminishing learning utility as activation variance stabilizes. Batches with $\Delta \bar{X} < \delta$ are pruned.

cern in data pruning is the risk of introducing bias by disproportionately removing informative samples or underrepresented classes. *B-PAS* operates at the batch level using activation stability and is agnostic to class labels and per-sample loss statistics; pruning decisions are therefore driven solely by model-internal dynamics rather than sample difficulty or data distribution. Across diverse experiments, *B-PAS* consistently preserves, and in some cases improves, classification accuracy compared to full-dataset training, indicating no measurable degradation in predictive performance. Furthermore, an analysis in Appendix A.6 shows that the pruning strategy does not favor either easy or hard examples and maintains class balance, resulting in a representative subset selected strictly according to the proposed criterion. Finally, our empirical analysis of pruning dynamics shows that batches gradually lose learning utility as training progresses and their activation variance stabilizes (Figure 5). Specifically, the change in mean standard deviation ($\Delta \bar{X}$) is initially large and decreases monotonically across epochs, eventually saturating. When this change falls below the threshold $\delta$, the corresponding batches are pruned. This mechanism enables *B-PAS* to eliminate redundant computation while prioritizing informative data. If activation stabilization were not indicative of reduced utility, pruning would significantly degrade performance; instead, our results demonstrate that activation stability serves as a reliable signal for scalable, efficient, and unbiased pruning. We conclude that *B-PAS* provides a practical, plug-and-play approach to data-efficient deep learning, with particular promise for large-scale training where efficiency gains are most impactful.

It is acknowledged that a current limitation is the use of empirically chosen threshold schedules; developing data-driven, adaptive thresholding mechanisms is an important direction for future work.

## ACKNOWLEDGMENTS

Research reported in this publication was supported by the Louisiana Board of Regents through the Board of Regents R & D, Research Competitiveness Subprogram (RCS) Support Fund under Contract Number: LEQSF(2023-26)-RD-A-23. Portions of this research were conducted with high-performance computational resources provided by the Louisiana Optical Network Infrastructure (http://www.loni.org).

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

# A  APPENDIX

## A.1  EXPERIMENTAL SETUP

### A.1.1  MODEL SPECIFICATIONS

**ResNet-18 and ResNet-50.** For CNN backbones, we adopt standard ResNet architectures with residual connections following (He et al., 2016). ResNet-18 is constructed using `BasicBlock` units (expansion factor 1) with layer configuration $[2, 2, 2, 2]$, while ResNet-50 uses `Bottleneck` units (expansion factor 4) with configuration $[3, 4, 6, 3]$. Both models begin with a $7 \times 7$ convolution and max pooling, followed by four residual stages, global average pooling, dropout (0.2), and a fully connected classifier. Batch Normalization is applied after each convolution, and ReLU serves as the activation. For activation tracking, we monitor outputs after each of the four residual stages (layer1–4).

**Convolutional vision Transformer (CvT).** We also evaluate *B-PAS* on CvT-13 (Wu et al., 2021), which integrates convolutional projections within transformer blocks. CvT-13 consists of three stages (Table 10): (i) a $64$-dim embedding with depth 1 and 1 attention head; (ii) a $192$-dim embedding with depth 2 and 3 heads; and (iii) a $384$-dim embedding with depth 10 and 6 heads. Each stage applies convolutional embedding, convolutional multi-head self-attention with depthwise projections, and MLP blocks with GELU activation. Activations are tracked at the stage outputs, where token sequences are reshaped back into spatial $(B, C, H, W)$ format for variance computation. The network ends with layer normalization, global average pooling over tokens, and a linear classifier.

Table 10: Model specifications for architectures used in our experiments.

| Model | Building Block | Depth | Embedding / Channels |
|---|---|---|---|
| ResNet-18 | BasicBlock | [2, 2, 2, 2] | [64, 128, 256, 512] |
| ResNet-50 | Bottleneck | [3, 4, 6, 3] | [256, 512, 1024, 2048] |
| CvT-13 | Conv-Attn + MLP | [1, 2, 10] | [64, 192, 384] |

***B-PAS* in GPT-2 Large.** We apply our batch pruning mechanism to GPT-2 Large (774M parameters) during instruction tuning on the Alpaca dataset, demonstrating generalization to autoregressive language models. **Activation Tracking:** We track activations at every alternate transformer decoder block throughout the model. During each forward pass, the hidden-state output of each tracked layer is captured, and we compute the standard deviation of the activations across layers. These per-layer statistics are then averaged to produce a single scalar summary representing the activation profile of each batch. **Pruning Decision:** After the first epoch, we compare the standard deviation of each batch's mean activation between consecutive epochs. If the absolute change in a batch's activation profile falls below a dynamically computed threshold, that batch is deemed converged as its activation patterns have stabilized, indicating diminishing marginal contribution to further learning. The threshold follows an exponential schedule that starts small and increases over training, allowing more aggressive pruning in later epochs as the model converges. Pruned batches are permanently removed from subsequent epochs, and the data loader is rebuilt at each epoch to reflect the reduced training set.

### A.1.2  DATASET SPECIFICATIONS

Table 11 provides the number of training and validation samples used for each dataset. CIFAR-10 and CIFAR-100 each include 50,000 training and 10,000 validation images. SVHN contains a larger validation set relative to its training size, with 73,257 training and 26,032 validation samples. ImageNet-1K, being significantly larger, includes over one million training images and 50,300 validation samples, reflecting its role as a large-scale benchmark. All images are augmented with commonly adopted transformations, i.e., normalization, random crop, and horizontal flip if not stated otherwise.

Table 11: Dataset Splits for Training and Validation

| Dataset | Training Samples | Validation Samples |
|---|---|---|
| CIFAR10 | 50,000 | 10,000 |
| CIFAR100 | 50,000 | 10,000 |
| SVHN | 73,257 | 26,032 |
| ImageNet-1K | 1,230,867 | 50,300 |

### A.1.3 HYPERPARAMETERS

Table 12 summarizes the hyperparameters used across datasets and architectures. Unless otherwise noted, all models are trained with SGD optimizer, using a momentum of 0.9. For ImageNet with ResNets, we adopt `MultiStepLR` scheduling, while other CNN datasets use cosine annealing. CvT models are trained with AdamW. For ImageNet training, we use 4 GPUs in parallel; hence, both the batch size and learning rate are scaled linearly by the number of GPUs (i.e., $256 \times 4$ total batch size and base learning rate $0.1 \times 4 = 0.4$). GPT-2 large is fine-tuned using the given hyperparameters.

Table 12: Training hyperparameters across datasets and architectures.

| Dataset / Model | Epochs | Batch Size | LR | Weight Decay | Scheduler |
|---|---|---|---|---|---|
| CIFAR-10 (ResNets) | 200 | 128 | 0.05 | $5 \times 10^{-4}$ | CosineAnnealing |
| CIFAR-100 (ResNets) | 200 | 128 | 0.10 | $5 \times 10^{-4}$ | CosineAnnealing |
| SVHN (ResNets) | 200 | 128 | 0.10 | $5 \times 10^{-4}$ | CosineAnnealing |
| ImageNet (ResNets) | 200 | 256×4 | 0.40 | $1 \times 10^{-4}$ | MultiStepLR |
| ImageNet (CvT) | 200 | 128×4 | $1 \times 10^{-3}$ | 0.05 | CosineAnnealing |
| GPT-2 large (fine-tuning) | 10 | 2×2 | $5 \times 10^{-5}$ | 0.01 | Linear |

### A.1.4 HARDWARE SPECIFICATIONS

Tables 13 and 14 summarize the computational setups. ImageNet experiments were performed on a high-performance cluster with dual 32-core Intel Xeon Platinum CPUs and $4\times$ A100 GPUs connected via NVLink. CIFAR-10/100 and SVHN experiments were conducted on a local workstation with an AMD Ryzen 9 CPU and a single Titan RTX GPU.

Table 13: Hardware specifications for CIFAR-10/100 and SVHN experiments.

| Component | Specification |
|---|---|
| CPU | AMD Ryzen 9 7900X, 12 cores / 24 threads, 4.7 GHz base |
| GPU | NVIDIA Titan RTX, 24 GB GDDR6 |
| RAM | Corsair Vengeance, 128 GB DDR5, 6000 MHz |

### A.1.5 SOFTWARE SPECIFICATIONS

Table 15 outlines the software environment used for all experiments. Python 3.8.18 served as the core programming language. Key libraries included PyTorch and Torchvision for model development, along with Matplotlib, NumPy, Scikit-learn, Seaborn, Pandas, and Pillow (PIL) for data handling and visualization. CIFAR-10/100 and SVHN experiments were conducted in Jupyter Notebook, facilitating interactive development and reproducibility. Additionally, ImageNet experiments were conducted as Python scripts.

Table 15: Software Specifications

| Component | Details |
|---|---|
| Python Version | 3.8.18 |
| Libraries | torch, torchvision, matplotlib, numpy, scikit-learn, seaborn, pandas, pillow (PIL) |
| Platform | Jupyter Notebook |

Table 14: Hardware specifications for ImageNet experiments.

| Component | Specification |
|---|---|
| CPU | Dual Intel Xeon Platinum 8358 (Ice Lake), 32 cores each |
| GPU | 4 × NVIDIA A100 (Ampere) with NVLink interconnect |
| RAM | 512 GB |

Table 16: Results for Group of $\delta_s = 5 \times 10^{-5}$

| $\delta_s$ | $\delta_e$ | DSI (%) | ACC (%) |
|---|---|---|---|
| $5 \times 10^{-5}$ | $1 \times 10^{-3}$ | 75 | 89.65 |
| $5 \times 10^{-5}$ | $5 \times 10^{-4}$ | 75 | 90.24 |
| $5 \times 10^{-5}$ | $1 \times 10^{-4}$ | 68 | 91.65 |

## A.2 MORE RELATED WORK

Bartoldson et al. (2020) analyzes weight pruning by defining stability as the accuracy drop after removing parameters, a diagnostic notion operating entirely in parameter space while keeping the data fixed. In contrast, *B-PAS* functions in data space, tracking temporal activation variance across epochs as an online signal for permanently pruning batches, a direction unexplored in prior pruning work. Similarly, Ganguli & Chong (2024) uses activation frequency to prune neurons in small fully connected networks, focusing on static model sparsification rather than data reduction. While activation patterns have been used to assess weight importance or characterize network behavior, no prior method leverages activation stability over time to directly remove training data.

## A.3 DETAILED TABULAR RESULTS

**Detailed CIFAR-10 Results.** Tables 16–20 provide the tabular counterpart of Figure 3, reporting the full results of our CIFAR-10 analysis across forty-five $(\delta_s, \delta_e)$ configurations. Each table corresponds to one starting threshold group, with multiple end thresholds. These results clearly illustrate the trade-off between pruning aggressiveness and accuracy:

- Larger $\delta_s$ values (Tables 16–17) trigger early and aggressive pruning, yielding substantial data savings but lower accuracy.
- Smaller $\delta_s$ values (Table 20) retain most data, preserving accuracy at the cost of reduced pruning benefits.
- Intermediate settings (Tables 18–19) achieve the most favorable balance, with the setting $\delta \in [10^{-6}, 5 \times 10^{-5}]$ (Table 19) delivering the highest accuracy while saving 25% of training data.

Together, these tables complement Figure 3 by providing a detailed numerical view of pruning dynamics, confirming that the thresholds govern an effective trade-off between efficiency and generalization.

**Detailed ImageNet Results.** Table 21 provides the tabular version of Figure 4, reporting results of *B-PAS* with ResNet-50 on ImageNet-1K across eight threshold settings. The table includes accuracy, Data Savings Index (DSI), training time, node-hours, and early stopping behavior, offering a more granular perspective on pruning dynamics.

A clear trade-off emerges between data savings and accuracy. Larger thresholds such as $\delta \in [5 \times 10^{-5}, 5 \times 10^{-4}]$ trigger aggressive pruning, with training terminating around epoch 100 and accuracy dropping to 76.1%, despite saving 73% node-hours. Conversely, smaller thresholds (e.g., $\delta \in [10^{-6}, 5 \times 10^{-5}]$) achieve slightly better accuracy compared to full dataset (78.63%) but achieve only moderate efficiency gains (37% node-hours saved).

Intermediate thresholds provide the most balanced trade-off: for instance, $\delta \in [5 \times 10^{-6}, 5 \times 10^{-5}]$ yields 78.43% accuracy with 47% DSI and 48% node-hours saved. GPU node-hours is calculated by (Training Time (in seconds)/3600)*4 (number of GPUs). The node-hours saved percentage is

Table 18: Results for Group of $\delta_s = 5 \times 10^{-6}$

| $\delta_s$ | $\delta_e$ | DSI(%) | ACC (%) |
|---|---|---|---|
| $5 \times 10^{-6}$ | $1 \times 10^{-3}$ | 57 | 92.28 |
| $5 \times 10^{-6}$ | $5 \times 10^{-4}$ | 55 | 92.97 |
| $5 \times 10^{-6}$ | $1 \times 10^{-4}$ | 45 | 94.46 |
| $5 \times 10^{-6}$ | $9 \times 10^{-5}$ | 45 | 94.20 |
| $5 \times 10^{-6}$ | $8 \times 10^{-5}$ | 43 | 94.28 |
| $5 \times 10^{-6}$ | $7 \times 10^{-5}$ | 40 | 94.81 |
| $5 \times 10^{-6}$ | $6 \times 10^{-5}$ | 41 | 94.71 |
| $5 \times 10^{-6}$ | $5 \times 10^{-5}$ | 40 | 94.59 |
| $5 \times 10^{-6}$ | $4 \times 10^{-5}$ | 37 | 94.79 |
| $5 \times 10^{-6}$ | $3 \times 10^{-5}$ | 31 | 95.02 |
| $5 \times 10^{-6}$ | $2 \times 10^{-5}$ | 28 | 95.05 |
| $5 \times 10^{-6}$ | $1 \times 10^{-5}$ | 23 | 95.44 |

Table 17: Results for Group of $\delta_s = 1 \times 10^{-5}$

| $\delta_s$ | $\delta_e$ | DSI(%) | ACC (%) |
|---|---|---|---|
| $1 \times 10^{-5}$ | $1 \times 10^{-3}$ | 63 | 91.28 |
| $1 \times 10^{-5}$ | $5 \times 10^{-4}$ | 59 | 92.28 |
| $1 \times 10^{-5}$ | $1 \times 10^{-4}$ | 52 | 93.61 |
| $1 \times 10^{-5}$ | $5 \times 10^{-5}$ | 47 | 94.31 |
| $1 \times 10^{-5}$ | $4 \times 10^{-5}$ | 43 | 94.49 |
| $1 \times 10^{-5}$ | $3 \times 10^{-5}$ | 43 | 94.58 |

Table 19: Results for Group of $\delta_s = 1 \times 10^{-6}$

| $\delta_s$ | $\delta_e$ | DSI(%) | ACC (%) |
|---|---|---|---|
| $1 \times 10^{-6}$ | $1 \times 10^{-3}$ | 47 | 93.24 |
| $1 \times 10^{-6}$ | $5 \times 10^{-4}$ | 42 | 94.13 |
| $1 \times 10^{-6}$ | $1 \times 10^{-4}$ | 31 | 95.18 |
| $1 \times 10^{-6}$ | $9 \times 10^{-5}$ | 29 | 95.04 |
| $1 \times 10^{-6}$ | $8 \times 10^{-5}$ | 30 | 94.99 |
| $1 \times 10^{-6}$ | $7 \times 10^{-5}$ | 29 | 95.07 |
| $1 \times 10^{-6}$ | $6 \times 10^{-5}$ | 26 | 95.44 |
| $1 \times 10^{-6}$ | $5 \times 10^{-5}$ | 25 | 95.60 |
| $1 \times 10^{-6}$ | $4 \times 10^{-5}$ | 22 | 95.47 |
| $1 \times 10^{-6}$ | $3 \times 10^{-5}$ | 22 | 95.28 |
| $1 \times 10^{-6}$ | $2 \times 10^{-5}$ | 16 | 95.54 |
| $1 \times 10^{-6}$ | $1 \times 10^{-5}$ | 15 | 95.49 |

Table 20: Results for Group of $\delta_s = 1 \times 10^{-7}$

| $\delta_s$ | $\delta_e$ | DSI (%) | ACC (%) |
|---|---|---|---|
| $1 \times 10^{-7}$ | $1 \times 10^{-3}$ | 34 | 94.47 |
| $1 \times 10^{-7}$ | $5 \times 10^{-4}$ | 31 | 94.94 |
| $1 \times 10^{-7}$ | $1 \times 10^{-4}$ | 18 | 95.49 |
| $1 \times 10^{-7}$ | $9 \times 10^{-5}$ | 19 | 95.46 |
| $1 \times 10^{-7}$ | $8 \times 10^{-5}$ | 19 | 95.50 |
| $1 \times 10^{-7}$ | $7 \times 10^{-5}$ | 18 | 95.33 |
| $1 \times 10^{-7}$ | $6 \times 10^{-5}$ | 15 | 95.52 |
| $1 \times 10^{-7}$ | $5 \times 10^{-5}$ | 14 | 95.43 |
| $1 \times 10^{-7}$ | $4 \times 10^{-5}$ | 12 | 95.57 |
| $1 \times 10^{-7}$ | $3 \times 10^{-5}$ | 11 | 95.55 |
| $1 \times 10^{-7}$ | $2 \times 10^{-5}$ | 9 | 95.60 |
| $1 \times 10^{-7}$ | $1 \times 10^{-5}$ | 5 | 95.60 |

calculated from the full dataset's node-hours (87.98). Importantly, DSI and node-hour savings follow consistent trends, reinforcing that pruning efficiency directly translates to a reduction in training costs. Early stopping occurs primarily under aggressive pruning settings, confirming that pruning not only reduces data usage but can also shorten training trajectories.

Table 21: Detailed ImageNet Results with node-hours, DSI, and Early Stopping.

| $\delta_s$ | $\delta_e$ | DSI(%) | Acc(%) | Early Stop Epoch | Training Time(s) | Node-hrs | Node-hrs Saved(%) |
|---|---|---|---|---|---|---|---|
| $1 \times 10^{-7}$ | $5 \times 10^{-4}$ | 38 | 78.46 | 178 | 46409.41 | 51.57 | 41 |
| $1 \times 10^{-6}$ | $5 \times 10^{-5}$ | 33 | 78.63 | – | 49855.24 | 55.39 | 37 |
| $5 \times 10^{-7}$ | $5 \times 10^{-4}$ | 46 | 78.55 | 175 | 41460.80 | 46.07 | 48 |
| $5 \times 10^{-6}$ | $5 \times 10^{-5}$ | 47 | 78.43 | – | 40919.18 | 45.47 | 48 |
| $5 \times 10^{-5}$ | $5 \times 10^{-4}$ | 74 | 76.10 | 102 | 21067.36 | 23.41 | 73 |
| $1 \times 10^{-5}$ | $5 \times 10^{-4}$ | 62 | 77.85 | 143 | 29016.79 | 32.24 | 63 |
| $1 \times 10^{-5}$ | $1 \times 10^{-4}$ | 57 | 78.07 | 179 | 30841.92 | 34.27 | 61 |
| $1 \times 10^{-5}$ | $5 \times 10^{-5}$ | 57 | 78.07 | 179 | 30828.36 | 34.25 | 61 |
| $5 \times 10^{-6}$ | $5 \times 10^{-5}$ | 37 | 71.50 | – | 29638.31 | 32.93 | 62 |
| Full Dataset | – | 0 | 78.07 | – | 79183.84 | 87.98 | 0 |

Table 22: Pattern of $\delta_s$–$\delta_e$ schedules on pruning (DSI) and accuracy under full-data vs. 10% of training data monitoring in CIFAR-10.

| $\delta_s$ | $\delta_e$ | DSI (Full) | Acc (Full) | DSI (10% data) | Acc (10% data) |
|---|---|---|---|---|---|
| $5 \times 10^{-5}$ | $5 \times 10^{-4}$ | 74.52 | 90.24 | 0.5655 | 79.0 |
| $10^{-5}$ | $5 \times 10^{-5}$ | 46.83 | 94.31 | 0.2528 | 81.8 |
| $5 \times 10^{-6}$ | $5 \times 10^{-5}$ | 39.78 | 94.59 | 0.1683 | 82.2 |
| $10^{-6}$ | $5 \times 10^{-5}$ | 25.06 | 95.60 | 0.1244 | 82.8 |
| $10^{-7}$ | $5 \times 10^{-5}$ | 14.26 | 95.43 | 0.1030 | 82.1 |

Table 23: Patttern of $\delta_s$–$\delta_e$ schedules under full-data vs. 10% of training data monitoring in ImageNet-1K.

| $\delta_s$ | $\delta_e$ | DSI (Full) | Acc (Full) | Time (Full) | DSI (10%) | Acc (10%) | Time (10%) |
|---|---|---|---|---|---|---|---|
| $10^{-7}$ | $5 \times 10^{-4}$ | 38.48 | 0.7846 | 46409.41 s | 34.71 | 0.5057 | 5274.02 s |
| $5 \times 10^{-6}$ | $5 \times 10^{-5}$ | 47.21 | 0.7843 | 40919.18 s | 43.16 | 50.4 | 4793.08 s |
| $10^{-5}$ | $5 \times 10^{-5}$ | 57.34 | 0.7807 | 30828.36 s | 51.78 | 50.14 | 4206.86 s |

## A.4 Fast and Reliable $\delta$ Selection Using a Small Subset of Training Data

The tables 22 and 23 show that $\delta_s$ and $\delta_e$ can be tuned quickly and reliably using only a small portion of the training set. Across both CIFAR-10 with ResNet-18 and ImageNet-1K with ResNet-50, the relative ordering of pruning strength and accuracy remains consistent between full data runs and ten percent subset runs. Larger thresholds such as $\delta_s = 10^{-5}$, $\delta_e = 5 \times 10^{-5}$ consistently yield higher DSI, while smaller thresholds such as $\delta_s = 10^{-7}$, $\delta_e = 5 \times 10^{-4}$ produce more conservative pruning, precisely matching the full training patterns. These partial runs are extremely lightweight, requiring only a few minutes for CIFAR-10 and roughly one hour for ImageNet-1K on four A-100 GPUs. As a result, selecting $\delta$ is fast, inexpensive, and does not diminish the overall efficiency gains of *B-PAS*.

Once a good $\delta$ schedule is identified for a dataset family, it transfers well to related settings. The values tuned on CIFAR-10 transfer directly to CIFAR-100 and SVHN without further adjustment, preserving the expected DSI and accuracy behavior in low-resolution CNNs. Similarly, values tuned on ImageNet-1K with ResNet-50 generalize to ImageNet-like datasets and other CNN variants. Although transformers exhibit slower and noisier activation stabilization, they can also be handled with a small subset of data, as demonstrated by our experiments with CvT-13 and GPT-2 large. These findings show that $\delta$ hyperparameters can be tuned rapidly on small data slices and reused across models, making *B-PAS* practical and scalable for new architectures and datasets.

## A.5 Extended Comparison with SOTA

Table 24 summarizes (Qin et al., 2024) the performance of representative data selection and pruning techniques on CIFAR-10 and CIFAR-100. Classical core-set and influence-based methods (e.g., Herding, Influence, K-Center) provide moderate gains, while more recent gradient- and uncertainty-based approaches (e.g., GraNd, EL2N, DP, UCB) achieve higher accuracy, especially on CIFAR-100. InfoBatch represents the prior state of the art, reaching 95.6% on CIFAR-10 and 78.2% on CIFAR-100. B-PAS matches these best-reported results, maintaining a better DSI despite using a fundamentally different criterion based on temporal activation stability rather than loss, gradient, or uncertainty-driven scoring. This highlights that activation variance dynamics can provide an equally strong or complementary signal for identifying redundant training data.

In addition, we compare the results from a recent large-scale pruning method: Large-scale Dataset Pruning with Dynamic Uncertainty (He et al., 2024). Their method achieves a 25% lossless pruning ratio on ImageNet-1K. By comparison, *B-PAS* removes up to 57% of the data while matching baseline accuracy and reducing GPU-hours by 61%, indicating substantially greater pruning capacity and compute savings on the same large-scale benchmark.

Table 24: Comparison of pruning methods on CIFAR-10 and CIFAR-100.

| Method | CIFAR-10 Acc. (%) | CIFAR-100 Acc. (%) |
|---|---|---|
| Herding (Welling, 2009) | 92.2 | 73.1 |
| Influence (Koh & Liang, 2017) | 93.1 | 74.4 |
| K-Center (Sener & Savarese, 2017) | 94.7 | 74.1 |
| DeepFool (Ducoffe & Precioso, 2018) | 95.1 | 74.2 |
| Forgetting (Toneva et al., 2018) | 94.7 | 75.3 |
| EL2N-2 (Paul et al., 2021) | 94.4 | 74.1 |
| EL2N-20 (Paul et al., 2021) | 95.3 | 77.2 |
| Least Confidence (Coleman et al., 2019) | 95.0 | 74.2 |
| Margin (Coleman et al., 2019) | 94.9 | 74.0 |
| CD (Agarwal et al., 2020) | 95.0 | 74.2 |
| Craig (Mirzasoleiman et al., 2020) | 94.8 | 74.4 |
| GraNd-4 (Paul et al., 2021) | 95.3 | 74.6 |
| Glister (Killamsetty et al., 2021) | 95.2 | 74.6 |
| DP (Yang et al., 2022) | 94.9 | 77.2 |
| $\varepsilon$-greedy (Raju et al., 2021) | 95.2 | 76.4 |
| UCB (Raju et al., 2021) | 95.3 | 77.3 |
| InfoBatch (Qin et al., 2024) | **95.6** | **78.2** |
| *B-PAS* (Ours) | **95.6** | **78.2** |

Table 25: Class distribution before training and after 100 epochs of *B-PAS*. The differences are within $\pm 0.1\%$, indicating no class imbalance introduced by pruning.

| Class (%) | plane | car | bird | cat | deer | dog | frog | horse | ship | truck |
|---|---|---|---|---|---|---|---|---|---|---|
| Initial | 10.00 | 10.00 | 10.00 | 10.00 | 10.00 | 10.00 | 10.00 | 10.00 | 10.00 | 10.00 |
| After 100 Epochs | 10.05 | 10.01 | 9.99 | 10.04 | 10.08 | 9.91 | 9.92 | 9.91 | 10.05 | 10.04 |

A.6 NATURE OF PRUNED BATCHES AND CLASS DISTRIBUTION

To examine whether *B-PAS* disproportionately removes "easy" examples, we conducted a difficulty analysis using two standard metrics from the pruning and curriculum learning literature: confidence (higher = easier) and misclassification rate (lower = easier). Confidence is computed as the maximum softmax probability for each sample, with higher values indicating that the model already finds the example easy. Misclassification rate reflects whether the model predicted the sample incorrectly at an early epoch, with lower values corresponding to easier examples. After pruning, these per-sample values are averaged over all samples contained within pruned batches and kept batches, so the reported means reflect the average difficulty of each group. As shown in Table 26, pruned batches have lower confidence (0.60 vs. 0.65) and higher misclassification rates (0.40 vs. 0.35) compared to kept batches.

If *B-PAS* were pruning easy samples, we would expect the opposite behavior. Instead, both metrics indicate that pruned batches are not easier, and the means are very close, demonstrating that *B-PAS* does not introduce difficulty bias and prunes based solely on activation stability rather than sample easiness.

Furthermore, to assess whether permanent pruning introduces class imbalance, we compared the class distribution of the dataset before training and after 100 epochs of *B-PAS*. As shown in Table 25, the

Table 26: Comparison of average difficulty metrics between pruned and kept batches on CIFAR-10 (ResNet-18). Pruned batches exhibit lower confidence and higher misclassification rates, indicating they are not disproportionately composed of "easy" examples.

| Metric | Pruned Mean | Kept Mean |
|---|---|---|
| Confidence | 0.60 | 0.65 |
| Misclassification Rate | 0.40 | 0.35 |

Table 27: Effect of different normalization strategies on DSI and accuracy.

| Normalizer | DSI (%) | Acc (%) |
|---|---|---|
| Layer | 13.2 | 93.99 |
| Layer + Batch | 21.9 | 94.97 |
| None | 2.0 | 90.39 |
| Batch | 25.06 | 95.60 |

proportions remain effectively unchanged, with all deviations within 0.1%. This stability occurs because *B-PAS* removes entire batches, and each batch contains a naturally mixed set of classes due to the initial random shuffling.

Consequently, pruning at the batch level does not preferentially remove any particular class. Combined with the difficulty analysis, these results demonstrate that *B-PAS* does not introduce class bias and does not prune based on sample easiness. Instead, pruning is driven solely by the activation-stability criterion, ensuring that the retained dataset remains representative of the original distribution with respect to both class balance and example difficulty.

Table 28: Comparison of Full Data training and *B-PAS* on CIFAR-10 (LeNet-5).

| Method | DSI (%) | Acc (%) |
|---|---|---|
| Full Data | – | 71.0 |
| *B-PAS* | 31.07 | 70.5 |

### A.7 MORE ABLATION STUDIES

**Extended Impact Analysis of Normalization.**

Table 27 shows that on CIFAR-10 with ResNet-18, BatchNorm yields the smoothest activation trajectories and therefore the highest data savings (25.06% DSI), but LayerNorm alone still enables meaningful pruning (13.2% DSI) with strong accuracy (93.99%). In contrast, removing normalization entirely severely destabilizes activation variance, leading to only 2% DSI and a substantial accuracy drop, which aligns with the known behavior of unnormalized ResNets.

Table 29: Effect of batch size on pruning effectiveness and accuracy on CIFAR-10 (ResNet-18).

| Batch Size | DSI (%) | Acc (%) |
|---|---|---|
| 32 | 20.25 | 94.86 |
| 64 | 23.26 | 94.93 |
| 128 | 25.06 | 95.60 |
| 256 | 26.97 | 95.10 |

The $\delta$-range ablation further confirms this trend in Table 30. Without BatchNorm, larger thresholds (e.g., $\delta_s = 5 \times 10^{-5}$, $\delta_e = 10^{-3}$) are required to counteract noisier activation dynamics, enabling 19.72% pruning at the cost of accuracy. More conservative thresholds (e.g., $\delta_s = 10^{-6}$, $\delta_e = 5 \times 10^{-5}$) yield minimal pruning (2%), reflecting the higher instability of unnormalized features. When BatchNorm is restored, the same thresholds ($10^{-6}$ to $5 \times 10^{-5}$) enable substantially higher DSI (25.06%) with top performance (95.60%), demonstrating that stable feature statistics directly expand the "prunable" region detected by *B-PAS*.

Table 30: Effect of pruning thresholds and BatchNorm on *B-PAS*.

| $\delta_s$ | $\delta_e$ | BatchNorm | DSI (%) | Accuracy (%) |
|---|---|---|---|---|
| $5 \times 10^{-5}$ | $10^{-3}$ | without BN | 19.72 | 89.87 |
| $10^{-6}$ | $5 \times 10^{-5}$ | without BN | 2.00 | 90.39 |
| $10^{-6}$ | $5 \times 10^{-5}$ | with BN | 25.06 | 95.60 |

Finally, Table 28 shows experiments on LeNet-5, a normalization-free architecture. It shows that *B-PAS* is not dependent on normalization layers. With a tuned $\delta$ schedule ($\delta_s = 10^{-4}$, $\delta_e = 10^{-3}$), the method prunes 31% of the data and maintains accuracy within 0.5% of the baseline. This confirms that *B-PAS* is compatible with models lacking normalization, but the threshold schedule must reflect the architecture's intrinsic activation stability regime.

**Effect of Batch Size.** The batch size ablation in Table 29 shows that DSI increases consistently as batch size grows, reflecting the fact that larger batches yield smoother and more stable activation trajectories across epochs, which allows *B-PAS* to prune earlier and more aggressively. Accuracy remains nearly unchanged across all settings, indicating that the method is robust to batch-size variation and does not introduce batch-size-dependent bias. Although larger batches improve the magnitude of achievable data savings, the core behavior of activation stabilization and accuracy preservation remains consistent, demonstrating that *B-PAS* functions reliably under standard training configurations.

Table 31: Comparison of different activation stability quantification for pruning.

| Quantification | DSI (%) | Acc (%) |
|---|---|---|
| kurtosis | 22.22 | 95.00 |
| entropy | 74.89 | 89.64 |
| max std | 13.45 | 95.41 |
| mean std | 25.06 | 95.60 |

**Different Approaches to Quantify Activation Stability.** The ablation in Table 31 compares several alternative activation stability metrics. Kurtosis yields moderate pruning but remains highly sensitive to early-epoch fluctuations, resulting in conservative DSI. Entropy exhibits the opposite behavior: its larger dynamic range causes overly aggressive pruning, leading to substantial accuracy degradation. Using the maximum standard deviation amplifies layer-wise noise and produces unstable pruning behavior with limited savings. In contrast, the proposed mean standard deviation provides a stable and well-behaved signal, achieving a balanced trade-off between pruning strength and accuracy. These results indicate that the mean standard deviation is the most reliable activation stability quantifier among the tested alternatives.

## A.8 VISUALIZATIONS

Figure 6 reports the number of batches pruned per epoch. Pruning does not occur in early epochs, when activation changes remain high, but becomes increasingly aggressive in later stages as changes stabilize.

Additionally, Figures 7 and 8 show the relationship between threshold growth and the number of remaining batches per epoch (CIFAR-10, ResNet-18). As training progresses, the threshold value becomes larger, resulting in more batch pruning.

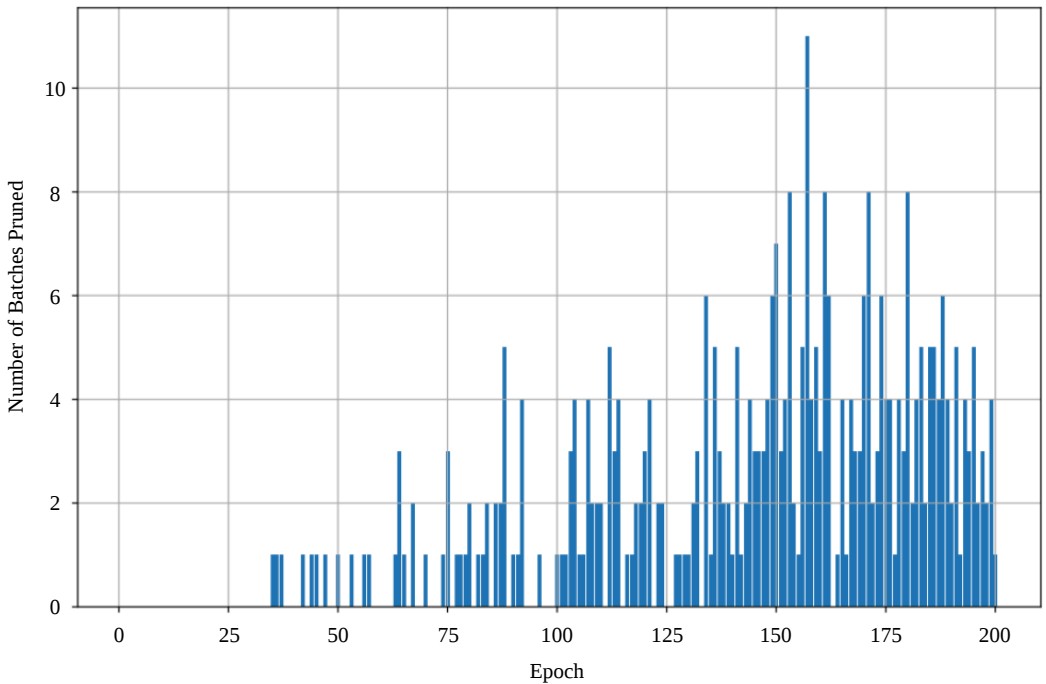

Figure 6: Number of batches pruned per epoch. No pruning occurs in early epochs, while pruning accelerates in later epochs as more batches stabilize, leading to substantial reductions in training cost.

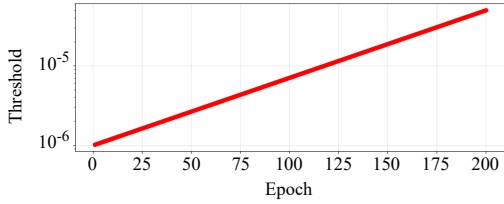

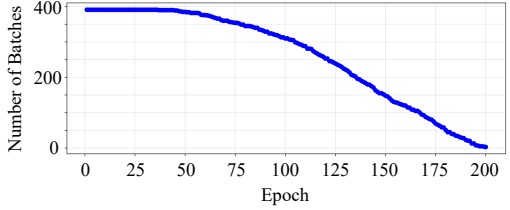

Figure 7: Exponential threshold evolution over epochs.

Figure 8: Remaining batches per epoch.

## A.9 LLM USAGE

In this research, large language models (LLMs) have been utilized to assist in verifying grammatical correctness. All contents were developed and verified by the authors.

