# OpenReview forum: "Batch Pruning by Activation Stability"
_ICLR.cc/2026/Conference — ICLR 2026 Poster_

### Official Review · Reviewer_ZSux · 2025-10-27

**Soundness:** 3
**Presentation:** 3
**Contribution:** 2
**Rating:** 4
**Confidence:** 3

**Summary:**

This paper introduces B-PAS (Batch Pruning by Activation Stability), a dynamic data pruning method designed to accelerate DNN training. The core idea is to permanently remove training batches whose internal activation statistics have stabilized across epochs, under the assumption that such batches contribute little to further learning. The method is presented as a lightweight, plug-and-play module. Experiments on various datasets and architectures show that B-PAS can reduce training data and GPU time while maintaining baseline accuracy.

**Strengths:**

1.  **A Simple and Effective Pruning Heuristic:** The paper proposes using activation stability as a signal for data pruning. This is a straightforward idea that leverages the model's internal state rather than relying on external metrics like loss. The "plug-and-play" nature of the module makes it easy to integrate into existing training pipelines.

2.  **Solid Efficiency Gains on Large-Scale Datasets:** The experiments demonstrate clear improvements in training efficiency, particularly on ImageNet. The method achieves a notable reduction in data usage and GPU hours compared to a full-dataset baseline, outperforming the InfoBatch method in terms of data savings while matching its accuracy.

**Weaknesses:**

1.  **High Sensitivity to Manually-Tuned Hyperparameters:** The method's performance is critically dependent on the `δ` threshold, which requires an extensive and costly grid search to tune (e.g., 45 configurations for CIFAR-10). The paper offers no automated way to set this parameter, which undermines its practical utility and the goal of saving computational resources.

2.  **Lack of Theoretical Insight:** The work is entirely empirical and lacks a theoretical foundation. It does not explain *why* activation stability is a reliable proxy for a batch's training utility. Without this analysis, the method remains a heuristic without a clear understanding of its potential failure modes.

3.  **Limited Baselines:** The comparison is mostly limited to InfoBatch. Other relevant data selection techniques like coreset selection are not discussed or compared against.

**Questions:**

1.  The notation in Section 2.2 for the second-to-last batch is `B_{n_i}-1`. This could be misinterpreted. Is the intended notation `B_{n_i-1}`? Please clarify.

2.  Given the high sensitivity to the δ schedule, have you explored any adaptive mechanisms for setting the threshold? For example, could δ be dynamically adjusted based on the validation set performance?

3.  The method's core metric is the mean standard deviation of activations. Have you investigated other metrics to quantify activation stability? For instance, using different statistical moments (e.g., kurtosis), information-theoretic measures (e.g., entropy), or different aggregation methods across layers (e.g., maximum vs. mean)? An ablation on this choice would strengthen the justification for the proposed metric.

4.  Regarding the non-monotonic accuracy curves in Figure 3: do you have a hypothesis for why a moderate increase in the pruning threshold can sometimes lead to slightly better generalization performance before it starts to degrade?

5.  The method relies on permanent pruning. Did you analyze the characteristics of the pruned batches? Is there a risk that this process could introduce bias by disproportionately removing samples from certain classes or domains?

---

> ### Author Response · Authors · 2025-11-19
> **Response to Reviewer ZSux (1/4)**
>
> We sincerely thank the reviewer for the thoughtful and insightful comments. We greatly appreciate the time and effort taken to provide such valuable feedback, which has significantly helped improve our work.
> ## Comment 1: Regarding Hyperparameter.
>
> Thank you for raising this point. We would like to clarify that the “45 settings” reported for CIFAR-10 were not an exhaustive grid search intended for deployment, but rather a research exploration to illustrate the full accuracy–DSI trade-off curve for both CIFAR-10 and ImageNet-1K. This was done solely to present a complete experimental landscape in the paper. We agree that in real applications, such an exhaustive sweep would be impractical, and we show below that it is not necessary.
>
> ### 1. Fast δ selection using only 10% of training data
>
> To evaluate practical tuning, we performed experiments using only 10% of the training data for rapid estimation. The δ–DSI–accuracy patterns remained consistent with the full training runs:
> CIFAR-10 (ResNet-18):
> | δₛ       | δₑ       | DSI (Full) | Acc (Full) | DSI (10% data) | Acc (10% data) |
> | -------- | -------- | ---------- | ---------- | ------------- | ------------- |
> | 5 × 10⁻⁵ | 5 × 10⁻⁴ | 74.52      | 90.24      | 0.5655        | 79.0          |
> | 1 × 10⁻⁵ | 5 × 10⁻⁵ | 46.83      | 94.31      | 0.2528        | 81.8          |
> | 5 × 10⁻⁶ | 5 × 10⁻⁵ | 39.78      | 94.59      | 0.1683        | 82.2          |
> | 1 × 10⁻⁶ | 5 × 10⁻⁵ | 25.06      | 95.6       | 0.1244        | 82.8          |
> | 1 × 10⁻⁷ | 5 × 10⁻⁵ | 14.26      | 95.43      | 0.1030        | 82.1          |
>
> ImageNet-1K (ResNet-50):
> | δₛ       | δₑ       | DSI (Full) | Acc (Full) | Training Time (s) (Full) | DSI (10% data) | Acc (10% data) | Training Time (s) (10% data) |
> | -------- | -------- | ---------- | ---------- | -------------------- | ------------- | ------------- | ----------------------- |
> | 1 × 10⁻⁷ | 5 × 10⁻⁴ | 38.48      | 0.7846     | 46409.41             | 34.71         | 0.5057        | 5274.02                 |
> | 5 × 10⁻⁶ | 5 × 10⁻⁵ | 47.21      | 0.7843     | 40919.18             | 43.16         | 50.4          | 4793.08                 |
> | 1 × 10⁻⁵ | 5 × 10⁻⁵ | 57.34      | 0.7807     | 30828.36             | 51.78         | 50.14         | 4206.86                 |
>
> The relative ranking and trade-off patterns remain intact, meaning δ can be chosen using only quick partial-training runs.
> In practice, these runs took only: ~1 hour for ImageNet-1K (10% subset on 4× A100 GPUs) and a few minutes for CIFAR-10
> Thus, δ tuning is fast, inexpensive, and does not offset efficiency gains.
>
> ### 2. Once δ is chosen for a dataset family, it transfers well
> Once δ is chosen for a dataset family, it tends to transfer well across similar datasets. For example, after identifying a suitable δ schedule for CIFAR-10, we used the same δ for CIFAR-100 and SVHN without any additional tuning. This resulted in the expected improvements in both DSI and accuracy, indicating that δ generalizes effectively across datasets within the same family.
> This observation suggests that δ is consistent across datasets and architecture families, particularly for low-resolution CNNs. Similarly, δ tuned for ImageNet-1K with ResNet-50 transfers well to other ImageNet-like datasets or other CNN variants.
> While transformers may require slightly adjusted δ schedules, these can be quickly identified by using a small subset of the data and leveraging intuition, just as we demonstrated with CvT and GPT-large.
>
> ### 3. Hyperparameter tuning is standard in deep learning, and δ behaves like any other hyperparameter
>
> We agree that tuning should be practical. However, hyperparameter tuning is a core and unavoidable part of deep learning. Hyperparameters (e.g., learning rate, batch size, number of epochs, dropout, layer width) are not learned by the model and usually set manually. Their optimal values vary across datasets, architectures, and training dynamics, and are typically selected via heuristics plus small-scale experiments. In short, Hyperparameter tuning is standard in deep learning, and δ schedules are no more burdensome than common tuning steps such as LR/weight-decay selection.
>
> Our δ parameters behave similarly to standard training hyperparameters. Their values are influenced by several factors, including the dataset’s difficulty and its convergence pattern. The type of architecture being used also plays a significant role, as does the presence or absence of normalization layers. Additionally, the length of the training process impacts the choice of δ, as longer training may require adjustments to ensure optimal performance.
>
> But tuning them does not require expensive searches. A small subset of training data is sufficient to choose a stable δ schedule.
> We thank the reviewer for highlighting this issue. We clarify hyperparameter selection guidelines and include this additional discussion in the Appendix A.3, Table 21 and 22 in the revised manuscript.

---

> ### Author Response · Authors · 2025-11-19
> **Response to Reviewer ZSux (2/4)**
>
> ## Comment 3: Extended SOTA comparison.
>
> Thank you for raising this point. We agree that broad comparison is important in pruning research. Our choice of InfoBatch as the main baseline is intentional and follows the precedent set by recent literature.
>
> ### Why InfoBatch is a strong and representative SOTA baseline.
> InfoBatch (Qin et al., 2024) conducted an extensive comparison against 16 prior state-of-the-art static and dynamic pruning methods and coreset selection, including Herding, Influence Functions, K-Center, DeepFool, Forgetting, EL2N, Craig, GraNd, Glister, ε-greedy, and UCB, covering nearly all prominent sample and dynamic pruning techniques. InfoBatch demonstrated superior accuracy under the same pruning ratios, establishing it as the strongest available benchmark. Because InfoBatch already outperforms these methods uniformly, using it as the primary baseline is consistent with prior work and ensures a fair, scalable comparison.
>
> ### Extended comparison for clarity.
> For completeness, we compiled the accuracy numbers reported for these methods on CIFAR-10 with ResNet-18 (30% pruning ratio):
>
>
> | Method                                   | CIFAR-10 Accuracy (%) | CIFAR-100 Accuracy (%) |
> |------------------------------------------|-------------------------|--------------------------|
> | Herding (Welling, 2009)                 | 92.2                   | 73.1                    |
> | Influence (Koh & Liang, 2017)           | 93.1                   | 74.4                    |
> | K-Center (Sener & Savarese, 2018)       | 94.7                   | 74.1                    |
> | DeepFool (Ducoffe & Precioso, 2018)     | 95.1                   | 74.2                    |
> | Forgetting (Toneva et al., 2018)        | 94.7                   | 75.3                    |
> | EL2N-2 (Toneva et al., 2018)            | 94.4                   | 74.1                    |
> | EL2N-20 (Toneva et al., 2018)           | 95.3                   | 77.2                    |
> | Least Confidence (Coleman et al., 2019) | 95.0                   | 74.2                    |
> | Margin (Coleman et al., 2019)           | 94.9                   | 74.0                    |
> | CD (Agarwal et al., 2020)               | 95.0                   | 74.2                    |
> | Craig (Mirzasoleiman et al., 2020)      | 94.8                   | 74.4                    |
> | GraNd-4 (Paul et al., 2021)             | 95.3                   | 74.6                    |
> | Glister (Killamsetty et al., 2021)     | 95.2                   | 74.6                    |
> | DP (Yang et al., 2023)                 | 94.9                   | 77.2                    |
> | ε-greedy (Raju et al., 2021)            | 95.2                   | 76.4                    |
> | UCB (Raju et al., 2021)                 | 95.3                   | 77.3                    |
> | InfoBatch (Qin et al., 2024)                 | **95.6**                    | **78.2**                   |
> | **B-PAS (Ours)**                         | **95.6**               | **78.2**                |
>
> As reported in InfoBatch, all of these methods achieve lower accuracy than InfoBatch under the same pruning ratio. For the 30% pruning setting specifically, InfoBatch obtains DSI values of 22% on CIFAR-10 and 19% on CIFAR-100, both of which are noticeably lower than the DSI achieved by B-PAS (25% and 24%, respectively).
> This distinction becomes even more pronounced on ImageNet-1K, where scalability is crucial. InfoBatch gets 28% data savings and 40% GPU-hours reduction while matching baseline accuracy, whereas B-PAS achieves up to 57% data savings and 61% GPU-hours reduction while maintaining the same accuracy and can even improve accuracy (+0.36%) under a more conservative threshold schedule.
>
> Since InfoBatch already outperforms the 16 prior static and dynamic pruning approaches, and B-PAS surpasses InfoBatch on both data efficiency and computational savings, the comparison is both representative and sufficient. The extended comparison table is included in Appendix A.3 as Table 23 in the revised paper.

---

> ### Author Response · Authors · 2025-11-19
> **Response to Reviewer ZSux (3/4)**
>
> ## Question 1: Notation clarification.
>
> In Section 2.2, the subscript
> i refers to the epoch index, not the batch index. Therefore:
>
> At epoch i, the batches are: B_{1}, B_{2},..., B_{n_i}. Where B_{n_i} is the last batch and B_{n_i}-1 denotes the second-to-last batch within the same epoch.
>
> At the previous epoch i-1, the batches are: B_{1}, B_{2},..., B_{n_{i-1}}. Where B_{n_{i-1}} is the last batch of epoch i-1 and B_{n_{i-1}}-1 is its second-to-last batch.
>
>
> ## Question 2: Adaptive mechanism for setting threshold.
>
> We explored several adaptive mechanisms for determining δ, including percentile-based thresholds, Gaussian-based statistical ranges, and moving-average or exponential-smoothing variants. In practice, these approaches remained heavily biased toward the large activation differences (shown in Figure 5) present in the early epochs. As a result, the pruning thresholds were dominated by early-epoch noise, causing batches to become dependent on these initial fluctuations rather than their true long-term stability. This led to overly aggressive pruning and substantial performance degradation. Since activation dynamics only stabilize later in training, early-epoch–driven adaptive methods proved unreliable. While adaptive, schedule-free pruning remains a promising direction for future work, our current results indicate that simple fixed δ-ranges are the most stable and effective as of now.
>
> ## Question 3: Activation stability quantification.
>
>
> Thank you for the suggestion. We evaluated several alternative ways of quantifying activation stability beyond the proposed mean standard deviation, including higher-order statistical moments, information-theoretic measures, and different aggregation strategies. For kurtosis, we computed the mean kurtosis of activations across layers and tracked its epoch-to-epoch change; thresholds were adjusted to match the natural kurtosis range [10⁻³, 5 × 10⁻²]. For entropy, we estimated the entropy of activation histograms for each batch (averaged across layers) and measured the change; since entropy values span a larger scale, the threshold schedule was set accordingly [10⁻³, 10⁻¹]. For max-std, we used the maximum standard deviation across layers instead of the mean, making the metric more sensitive to layer-wise noise. These variants produced the following results on CIFAR-10 with ResNet-18:
> | Quantification | DSI (%)   | Acc (%) |
> |----------------|-------|---------|
> | kurtosis       | 22.22 | 95.00   |
> | entropy        | 74.89 | 89.64   |
> | max std        | 13.45 | 95.41   |
> | mean std            | 25.06 | 95.60   |
>
> Kurtosis and entropy are highly sensitive to early-epoch activation fluctuations, which cause either conservative pruning (kurtosis) or aggressive pruning with accuracy loss (entropy). Max-std leads to reduced DSI due to considering the maximum value. Overall, mean standard deviation consistently provides the most stable signal, balancing pruning strength and accuracy preservation. This result is added in Appendix A.4, Table 31 in the revised paper.

---

> ### Author Response · Authors · 2025-11-19
> **Response to Reviewer ZSux (4/4)**
>
> ## Question 4: Non-monotonic accuracy curves in Figure 3 hypothesis
> Thank you for the question. We believe the slight non-monotonicity in the CIFAR-10 accuracy curve arises from the small scale and relatively low complexity of the dataset. CIFAR-10 often exhibits mild fluctuations in generalization when moderate regularization is introduced (e.g., dropout, data augmentation intensity, mixup strength), and B-PAS behaves similarly: a slightly stronger pruning threshold can act as a mild regularizer, occasionally improving generalization before accuracy begins to decline once pruning becomes too aggressive.
>
> However, this effect is much less pronounced on ImageNet-1K, as shown in Figure 4. On large-scale datasets, the accuracy–DSI trade-off becomes smoother, clearly reflecting the expected behavior: gentle pruning maintains accuracy, while aggressive pruning eventually degrades it. Thus, we interpret the CIFAR-10 irregularities as artifacts of small-dataset variability rather than a core property of the method.
>
> ## Question 5: Characteristics of the pruned batches
>
> We performed a detailed analysis to examine whether B-PAS disproportionately prunes “easy” examples on CIFAR-10 (ResNet-18). Following standard definitions in the pruning and curriculum learning literature, we evaluated difficulty using two widely accepted metrics: confidence and misclassification rate.
>
> ### What do the difficulty metrics mean and how they are computed
>
> (a) Confidence (higher = easier)
>
> (i) Computed as: confidence = max(softmax logits)
>
> (ii) If a model predicts an example with very high confidence, it is typically considered “easy.”
>
> (iii) Lower confidence implies more uncertainty and typically indicates a “harder” example.
>
> (b) Misclassification Rate (lower = easier)
>
> (i) Computed by checking whether the model misclassified each sample at an early epoch.
>
> (ii) A low misclassification rate indicates the model already finds these examples easy.
>
> (iii) Higher misclassification occurs for harder examples.
>
> Once pruning occurs, these per-sample values are averaged within the pruned batches and within the kept batches, producing the mean values shown below.
> | Metric                 | Pruned Batches Mean | Kept Batches Mean |
> |------------------------|---------------------|--------------------|
> | Confidence             | 0.60                | 0.65               |
> | Misclassification Rate | 0.40                | 0.35               |
>
> Pruned batches show lower confidence (0.60 vs. 0.65) and higher misclassification rates (0.40 vs. 0.35) compared to kept batches. If B-PAS were pruning easy examples, we would expect the opposite: higher confidence and fewer misclassifications in the pruned set. Instead, both metrics indicate that pruned batches are not easier. The means are also close, showing no difficulty bias.
>
>
> Furthermore, to check whether pruning introduces any class imbalance, we measured the class distribution before training and after 100 epochs of B-PAS on CIFAR-10 (ResNet-18). The distribution remains virtually unchanged:
>
> |                | plane | car  | bird | cat  | deer | dog  | frog | horse | ship | truck |
> |-----------------------|-------|------|------|------|------|------|------|-------|------|--------|
> | Initial (%)           | 10.00 | 10.00 | 10.00 | 10.00 | 10.00 | 10.00 | 10.00 | 10.00 | 10.00 | 10.00 |
> | After 100 Epochs (%)  | 10.05 | 10.01 | 9.99  | 10.04 | 10.08 | 9.91  | 9.92  | 9.91  | 10.05 | 10.04 |
>
> These differences are within ±0.1%, indicating no meaningful shift in class proportions. Because B-PAS prunes at the batch level and batches contain mixed classes, the class distribution of the retained data remains effectively identical to the original dataset.
>
> These results confirm that B-PAS does not prune based on example easiness, but strictly based on activation stability, independent of class or difficulty. So, there is no risk that this process could introduce bias by disproportionately removing samples from certain classes.
>
> The results are included in Appendix A.3, Table 25 and 26 in the revised paper.

---

> ### Comment · Reviewer_ZSux · 2025-11-24
> **Acknowledgment of Rebuttal**
>
> I thank the authors for their detailed and comprehensive response. I appreciate the effort put into conducting additional experiments and clarifying the methodology.
> * Regarding Hyperparameter Tuning: The demonstration that δ can be effectively tuned using only 10% of the training data (and that the relative ranking of configurations is consistent with full training) significantly alleviates my concern regarding the computational cost of finding the optimal threshold. This makes the method much more practical.
> * Regarding Baselines: I accept the argument that since InfoBatch has already been shown to outperform a wide range of static and dynamic pruning methods (Coreset, GraNd, etc.), comparing primarily against InfoBatch is sufficient to establish SOTA performance. The addition of Table 23 in the Appendix provides the necessary context.
> * Regarding Pruned Batch Characteristics: The analysis regarding the "difficulty" of pruned batches and the class distribution is insightful. It is reassuring to see that the method relies on activation stability rather than simply discarding "easy" samples, and that it maintains class balance, mitigating concerns about potential bias.
> * Regarding Metrics and Adaptive Thresholds: The ablation study on different stability metrics (kurtosis, entropy, etc.) and the explanation regarding the failure of early-epoch adaptive thresholds provide a better empirical justification for the chosen design.
>
> Overall, the rebuttal has successfully addressed my major concerns regarding the practicality of the method and the robustness of the evaluation. The method is simple, effective, and now shown to be practically tunable.
>
> Consequently, I am raising my score to 6.

---

> ### Author Response · Authors · 2025-11-26
>
> Dear reviewer ZSux,
>
> Thank you very much for your thoughtful follow-up and for taking the time to evaluate our rebuttal and additional experiments carefully. We are glad that the clarifications on the practicality of hyperparameter tuning, baseline justification, pruned-batch characteristics, and alternative metrics addressed your concerns. We sincerely appreciate your positive assessment of the method’s simplicity, effectiveness, and practical tunability, as well as your decision to raise the score. Your feedback has helped us strengthen the paper.

---

### Official Review · Reviewer_FtHA · 2025-10-31

**Soundness:** 2
**Presentation:** 3
**Contribution:** 2
**Rating:** 4
**Confidence:** 3

**Summary:**

To reduce the training costs of large-scale models, people have considered to reduce the number of training iterations via data pruning. This work proposes a data pruning method which dynamically prunes batches with small changes in activation variances across successive training epochs. Empirical results show that the proposed method can achieve better training efficiency without accuracy loss.

**Strengths:**

1. This work proposes to assess the contribution of batches using changes in their activation variances, which directly utilizes activation outputs from the forward pass, requiring no substantial additional computation, making it simple and efficient.

2. The paper is well written and easy to follow, clearly describing the motivation of the work and the proposed algorithm.

**Weaknesses:**

1. In line 53, the paper criticizes the existing methods, saying that they often rely on complex heuristics. However, the proposed methods also introduce additional hyperparameters that need to be tuned.

2. In the experiments, only one baseline (i.e., InfoBatch) is compared with the proposed method, which reduces the significance of the empirical results. Other recent data pruning algorithms, such as (He et al, 2024), are not compared.

3. Intuitively, the performance of the proposed method and the selection of the hyperparameters are likely to be greatly influenced by the batch size. However, this important factor was not considered in the ablation study.

**Questions:**

The proposed method is only tested on image classification tasks. Can it be generalized to other tasks (e.g., text generation) or models with a larger scale?

---

> ### Author Response · Authors · 2025-11-19
> **Response to Reviewer FtHA (1/2)**
>
> We thank the reviewer for the insightful comments. We greatly appreciate the time and effort taken to provide such valuable feedback.
> ## Comment 1: Regarding Hyperparameters.
>
> Our intention was not to claim that B-PAS is free of hyperparameters, but rather that it avoids complex, model-specific heuristics such as influence estimation, gradient-to-loss ratios, forgetting event thresholds, or sample-wise scoring functions used in prior methods. B-PAS introduces only two scalar hyperparameters that control the activation-stability range. As shown in our hyperparameter analysis in Appendix A.3, Table 21 and 22 in the revised manuscript, these can be selected quickly using a small subset (10% of data) without the need for an exhaustive search, and once chosen, they transfer well across similar datasets and architectures (e.g., CIFAR-10 → CIFAR-100/SVHN).
> Thus, while B-PAS does require setting thresholds similar to learning rate or weight decay in standard training, the tuning process is lightweight and is practical in real settings.
>
>
> ## Comment 3: Batch size ablation.
>
> We have added an ablation study on batch size (CIFAR-10, ResNet-18) for default CIFAR-10 δ-range choices to evaluate its impact on B-PAS. The results are shown below:
> | Batch Size | DSI (%)   | Acc (%) |
> |------------|-------|----------|
> | 32         | 20.25 | 94.86    |
> | 64         | 23.26 | 94.93    |
> | 128        | 25.06 | 95.60    |
> | 256        | 26.97 | 95.10    |
>
> We observe that DSI increases smoothly with batch size, as larger batches naturally produce more stable activation patterns across epochs, enabling earlier pruning. Importantly, accuracy remains stable across all batch sizes, indicating that B-PAS is not sensitive to batch-size-induced bias and remains effective under standard training configurations.
>
> While batch size influences the magnitude of achievable data savings, the overall behavior of the method, including activation stability trends and accuracy preservation, remains consistent. This ablation is included in Appendix A.4, Table 30 in the revised manuscript and thank you for highlighting this important factor.
>
>
> ## Question 1: Applying B-PAS to LLMs.
> Yes, B-PAS can be generalized to other tasks and has been tested on large language models. As a demonstration, we applied B-PAS to fine-tune GPT2-large on the Alpaca instruction-tuning dataset using a more aggressive threshold range (δₛ = 10⁻³
> , δₑ = 10⁻²), which is necessary given the higher complexity and smoother activation dynamics of transformer-based LLMs. The results (now included in Appendix A.3, Table 24 in the revised paper) are as follows:
> | Method     | Loss   | Perplexity | Avg. Epoch Time (s) | Total Time (s)  | Pruned Batch (%) | DSI (%)  |
> |------------|--------|------------|------------------|-------------|-------------------|------|
> | Full Data  | 0.2207 | 1.25       | 5359.88          | 54420.13    | 0                 | 0    |
> | B-PAS     | 0.2201 | 1.25       | 5039.11          | 51211.29    | 23.00%            | 6 |
>
> Since this is a small-scale fine-tuning task (10 epochs), activation stabilization happens later in training, which is why DSI is lower than in our previous large-scale results. This is consistent with our findings in Table 6 of the paper: longer training yields higher DSI, as batches require more time to reach activation stability. Nevertheless, even in this short training setting, B-PAS pruned 23% of batches, matched the baseline perplexity, slightly improved loss, and reduced total training time by about an hour on 2× NVIDIA A100 GPUs, demonstrating practical computational savings.
> Overall, this experiment shows that B-PAS is compatible with transformer-based LLMs and can provide measurable efficiency gains even in smaller fine-tuning scenarios. We added these findings in the revised paper (line 482-483).

---

> ### Author Response · Authors · 2025-11-19
> **Response to Reviewer FtHA (2/2)**
>
> ## Comment 2: Extended SOTA comparison.
>
> Thank you for raising this point. We agree that broad comparison is important in pruning research. Our choice of InfoBatch as the main baseline is intentional and follows the precedent set by recent literature.
>
> ### Why InfoBatch is a strong and representative SOTA baseline.
> InfoBatch (Qin et al., 2024) conducted an extensive comparison against 16 prior state-of-the-art static and dynamic pruning methods, including Herding, Influence Functions, K-Center, DeepFool, Forgetting, EL2N, Craig, GraNd, Glister, ε-greedy, and UCB, covering nearly all prominent sample and dynamic pruning techniques. InfoBatch demonstrated superior accuracy under the same pruning ratios, establishing it as the strongest available benchmark. Because InfoBatch already outperforms these methods uniformly, using it as the primary baseline is consistent with prior work and ensures a fair, scalable comparison.
>
> ### Extended comparison for clarity.
> For completeness, we compiled the accuracy numbers reported for these methods on CIFAR-10 with ResNet-18 (30% pruning ratio):
>
>
> | Method                                   | CIFAR-10 Accuracy (%) | CIFAR-100 Accuracy (%) |
> |------------------------------------------|-------------------------|--------------------------|
> | Herding (Welling, 2009)                 | 92.2                   | 73.1                    |
> | Influence (Koh & Liang, 2017)           | 93.1                   | 74.4                    |
> | K-Center (Sener & Savarese, 2018)       | 94.7                   | 74.1                    |
> | DeepFool (Ducoffe & Precioso, 2018)     | 95.1                   | 74.2                    |
> | Forgetting (Toneva et al., 2018)        | 94.7                   | 75.3                    |
> | EL2N-2 (Toneva et al., 2018)            | 94.4                   | 74.1                    |
> | EL2N-20 (Toneva et al., 2018)           | 95.3                   | 77.2                    |
> | Least Confidence (Coleman et al., 2019) | 95.0                   | 74.2                    |
> | Margin (Coleman et al., 2019)           | 94.9                   | 74.0                    |
> | CD (Agarwal et al., 2020)               | 95.0                   | 74.2                    |
> | Craig (Mirzasoleiman et al., 2020)      | 94.8                   | 74.4                    |
> | GraNd-4 (Paul et al., 2021)             | 95.3                   | 74.6                    |
> | Glister (Killamsetty et al., 2021)     | 95.2                   | 74.6                    |
> | DP (Yang et al., 2023)                 | 94.9                   | 77.2                    |
> | ε-greedy (Raju et al., 2021)            | 95.2                   | 76.4                    |
> | UCB (Raju et al., 2021)                 | 95.3                   | 77.3                    |
> | InfoBatch (Qin et al., 2024)                 | **95.6**                    | **78.2**                   |
> | **B-PAS (Ours)**                         | **95.6**               | **78.2**                |
>
> As reported in InfoBatch, all of these methods achieve lower accuracy than InfoBatch under the same pruning ratio. For the 30% pruning setting specifically, InfoBatch obtains DSI values of 22% on CIFAR-10 and 19% on CIFAR-100, both of which are noticeably lower than the DSI achieved by B-PAS (25% and 24%, respectively).
> This distinction becomes even more pronounced on ImageNet-1K, where scalability is crucial. InfoBatch gets 28% data savings and 40% GPU-hours reduction while matching baseline accuracy, whereas B-PAS achieves up to 57% data savings and 61% GPU-hours reduction while maintaining the same accuracy and can even improve accuracy (+0.36%) under a more conservative threshold schedule.
>
> Since InfoBatch already outperforms the 16 prior static and dynamic pruning approaches, and B-PAS surpasses InfoBatch on both data efficiency and computational savings, the comparison is both representative and sufficient.
>
> ### Additional SOTA: Large-scale Dataset Pruning with Dynamic Uncertainty (He et al., 2024).
> We also considered the recent method by He et al. (2024) titled “Large-scale Dataset Pruning with Dynamic Uncertainty.” On ImageNet-1K, their method reports a 25% lossless pruning ratio. In contrast, B-PAS achieves up to 57% pruning while matching baseline accuracy and reducing GPU-hours by 61%, demonstrating significantly higher pruning capacity and computational savings on the same large-scale dataset.
>
> The extended comparison table is included in Appendix A.3 in the revised paper.

---

> > ### Comment · Reviewer_FtHA · 2025-11-28
> >
> > Thanks for the detailed responses. I still have some doubts regarding the ablation study on batch size. As the authors claimed in the response, larger batches naturally yield more stable activation patterns across epochs, enabling earlier pruning. However, intuitively, pruning a large batch of data early might negatively affect the model performance. Yet the current results show that as batch size increases, DSI incresases while accuracy remains stable or even improves slightly. How can this be explained?

---

> ### Author Response · Authors · 2025-11-28
> **Clarification of the ablation study on batch size**
>
> Thank you for the follow-up question. We appreciate the opportunity to clarify this behavior. The current results show that as the batch size increases, DSI increases, but after a point, accuracy starts to degrade. In this regime, pruning a large batch of data is negatively affecting the model performance. The trend in the table (the same table in the previous response and Table 30 in the updated paper) aligns with well-established theory on batch size, SGD noise, and generalization behavior observed in prior work [1] [2] [3] [4], and the interaction with pruning becomes easier to interpret once these dynamics are taken into account.
>
> Ablation Study on Batch Size:
>
> | Batch Size | DSI (%)   | Acc (%) |
> |------------|-------|----------|
> | 32         | 20.25 | 94.86    |
> | 64         | 23.26 | 94.93    |
> | 128        | 25.06 | 95.60    |
> | 256        | 26.97 | 95.10    |
>
> First, the increase in accuracy from batch size 32 to 128 is expected. Moderate increases in batch size generally stabilize the optimization process when the learning-rate–batch-size ratio remains in a good regime (as characterized by [3]). Larger batches reduce gradient noise, producing smoother activation dynamics and more consistent gradient updates. For CIFAR-10 specifically, [2] also shows that increasing the batch size to moderate levels can even improve generalization. This aligns with our results: batch size 128 achieves the highest accuracy (95.60%) among the four settings.
>
> However, increasing batch size beyond this point begins to introduce the well-known generalization gap associated with large-batch training [1] [4]. Because we use an unchanged learning rate of 0.05 across all settings of CIFAR-10, pushing the batch size to 256 begins to move the training dynamics out of the optimal noise regime, causing the small accuracy drop observed in the table. Importantly, this degradation occurs even without pruning in prior studies, so its presence here is expected.
>
> Second, the monotonic rise in DSI with batch size follows naturally from the mechanics of B-PAS. Larger batches average over more samples, producing smoother activation trajectories and faster stabilization. This causes earlier pruning and a higher DSI. This behavior is not detrimental by itself; it simply reflects that stability emerges sooner when batch-level activation statistics are computed over more data.
>
> Finally, the mild accuracy drop at batch size 256 can be attributed to the inherent generalization gap of large-batch SGD under fixed learning-rate schedules. These effects explain why accuracy peaks at 128, where stability, gradient noise, and pruning selectivity are best balanced, and then declines slightly at 256 despite the higher DSI.
>
> In summary, small batches yield unstable training and lower DSI; moderate batches (≈128) provide the best accuracy and balanced pruning; and very large batches (>128) naturally exhibit both a generalization gap and more aggressive pruning, leading to the mild accuracy decrease observed. This behavior is consistent with prior theoretical and empirical findings on batch size.
>
> ## References
>
> [1] Hoffer, E., Hubara, I., & Soudry, D. (2017). Train longer, generalize better: closing the generalization gap in large batch training of neural networks. Advances in neural information processing systems, 30.
>
> [2] Smith, S. L., Kindermans, P. J., Ying, C., & Le, Q. V. (2018, February). Don't Decay the Learning Rate, Increase the Batch Size. In International Conference on Learning Representations.
>
> [3] He, F., Liu, T., & Tao, D. (2019). Control batch size and learning rate to generalize well: Theoretical and empirical evidence. Advances in neural information processing systems, 32.
>
> [4] Keskar, N. S., Mudigere, D., Nocedal, J., Smelyanskiy, M., & Tang, P. T. P. (2017, February). On Large-Batch Training for Deep Learning: Generalization Gap and Sharp Minima. In International Conference on Learning Representations.

---

### Official Review · Reviewer_BNgi · 2025-11-02

**Soundness:** 3
**Presentation:** 4
**Contribution:** 3
**Rating:** 6
**Confidence:** 4

**Summary:**

This paper introduces "Batch Pruning by Activation Stability" (B-PAS), a novel, dynamic data pruning strategy to accelerate deep neural network training. The method operates as a lightweight, plug-and-play module that permanently removes batches from the training process if their contribution to learning has diminished. The core mechanism leverages a simple, model-internal signal: the stability of post-ReLU activation feature maps. By tracking the change in the mean standard deviation of activations for each batch across consecutive epochs, B-PAS prunes batches where this change falls below a dynamic threshold. The authors provide comprehensive empirical validation on various models (ResNet, CvT) and datasets (CIFAR, ImageNet), demonstrating up to 57% data savings and a 61% reduction in GPU node-hours on ImageNet without sacrificing accuracy, significantly outperforming prior art. The paper also introduces the Data Savings Index (DSI), a hardware-agnostic metric for data efficiency.

**Strengths:**

Novelty and Simplicity: The core idea of using the stability of post-ReLU activation variance as a pruning signal is original, computationally cheap, and mechanistically sound. It provides an elegant, model-internal alternative to more complex methods that rely on loss or gradient statistics.

Great Empirical Rigor: The method is thoroughly validated across diverse architectures (ResNet, CvT) and datasets (CIFAR, SVHN, ImageNet-1K). The comprehensive ablation studies and the significant outperformance of a strong baseline (InfoBatch) on ImageNet provide convincing evidence of the method's quality and effectiveness.

Significant Practical Impact: The demonstrated efficiency gains are substantial, with up to a 61% reduction in GPU node-hours on ImageNet without accuracy loss. The introduction of the hardware-agnostic Data Savings Index (DSI) is also a valuable contribution to the community for standardizing efficiency reporting.

**Weaknesses:**

Critical Dependency on Batch Normalization: The method's effectiveness collapses without Batch Normalization, as shown in the ablation study (DSI drops from 25% to 2%). This severely limits its applicability to the growing number of modern architectures, such as many transformers, that do not use BN.

Reduced Effectiveness on Transformers: B-PAS achieves significantly lower data savings on the Convolutional Vision Transformer (CvT) compared to CNNs (14% vs. 47% DSI on ImageNet). This suggests the pruning signal is architecture-dependent and questions the method's generalizability beyond CNNs.

Practicality Concerns of Hyperparameter Tuning: The method's performance is highly sensitive to the pruning threshold schedule ($\delta_s, \delta_e$), requiring extensive grid searches (e.g., 45 settings for CIFAR-10) to find the optimal configuration. This tuning overhead could offset the efficiency gains in practice.

Deviation from Standard Training Practices: The requirement to fix batch compositions at initialization to enable tracking is a departure from the standard practice of reshuffling the dataset each epoch. This could have unexamined negative effects on model generalization.

**Questions:**

On Fixed Batch Composition: Since using fixed batches deviates from standard shuffling, what is the impact on generalization? Does intra-batch shuffling fully mitigate potential overfitting to specific sample groupings?

On Batch Normalization Dependency: Given the method's strong reliance on Batch Normalization, does the stability signal depend on BN's per-batch standardization? Have you tested its effectiveness with other schemes like LayerNorm or in Normalizer-Free architectures?

On Transformer Performance: Could you elaborate on why B-PAS is less effective on the CvT architecture? Is the activation stability signal inherently weaker in transformers, and did you investigate tracking activations at different points within the transformer blocks?

On Hyperparameter Sensitivity: The $(\delta_s, \delta_e)$ schedule required extensive tuning. Are there any practical heuristics for setting these hyperparameters for new tasks without performing a costly grid search?

On the Nature of Pruned Batches: Could you provide a qualitative analysis of the pruned batches? For instance, are they primarily "easy" examples, and does the pruning process introduce any significant shift in the class distribution of the remaining data?

---

> ### Author Response · Authors · 2025-11-19
> **Response to Reviewer BNgi (1/4)**
>
> We sincerely thank the reviewer for the thoughtful and insightful comments. We greatly appreciate the time and effort taken to provide such valuable feedback, which has significantly helped improve our work.
> ## Comment 1 and Question 2: Regarding Batch Normalization.
>
> We agree that understanding how B-PAS behaves with different normalization schemes is crucial for assessing its generality. We have expanded our ablation studies accordingly and added the results to the appendix.
>
>
>
> ### 1. Updated comparison across normalization schemes.
>
> We conducted experiments with LayerNorm, BatchNorm, LayerNorm+BatchNorm, and no normalization using ResNet-18 on CIFAR-10. The results are:
> | Normalizer     | DSI (%)  | Acc (%) |
> |----------------|-------|---------|
> | LayerNorm          | 13.2  | 93.99   |
> | LayerNorm + BatchNorm  | 21.9  | 94.97   |
> | None           | 2.0   | 90.39   |
> | BatchNorm          | 25.06 | 95.60   |
>
> These results show that while BatchNorm offers the smoothest activation dynamics (and thus the highest DSI), LayerNorm also supports meaningful pruning, achieving 13.2% DSI with strong accuracy. The “None” case is indeed challenging due to unstable activation variance and gradient explosion, which is a known property of unnormalized ResNets. We apologize for not including this extended study in the original submission.
>
> ### 2. Impact of varying δ ranges without BatchNormalization.
>
> We also revisited the ablation using different δ settings, and Table 5 in the revised paper is updated according to this:
>
> | δₛ       | δₑ       | BatchNorm  | DSI (%) | Accuracy (%) |
> | -------- | -------- | ---------- | ------- | ------------ |
> | 5 × 10⁻⁵ | 10⁻³ | without BN | 19.72   | 89.87        |
> | 10⁻⁶ | 5 × 10⁻⁵ | without BN | 2       | 90.39        |
> | 10⁻⁶ | 5 × 10⁻⁵ | with BN    | 25.06   | 95.60        |
>
>
> This demonstrates that B-PAS remains effective without BN, but requires more aggressive threshold ranges to compensate for the noisier activation trajectories. In architectures where activation dynamics are less smooth (such as unnormalized ResNets), stable pruning requires recalibrated δ schedules; this is expected, as BN is specifically designed to prevent gradient explosion and stabilize feature statistics.
>
> ### 3. Applicability to Normalizer-Free Architectures.
>
> To further evaluate this concern, we tested B-PAS on LeNet-5, a normalization-free architecture:
> | Method    | DSI   | Acc (%) |
> |-----------|-------|----------|
> | Full Data | 0     | 71.0     |
> | B-PAS     | 31.07 | 70.5     |
>
> With threshold values (δₛ = 10⁻⁴, δₑ = 10⁻³), B-PAS pruned 31% of batches while maintaining accuracy within 0.5%, indicating that the method does work in normalization-free networks, but, as expected, requires appropriately tuned δ to match the activation stability regime of the architecture.
>
> Furthermore, we applied B-PAS to fine-tune GPT2-large, a modern, widely used transformer architecture that does not use normalization on the Alpaca instruction-tuning dataset using a more aggressive threshold range (δₛ = 10⁻³, δₑ = 10⁻²), which is necessary given the higher complexity and smoother activation dynamics of transformer-based LLMs. The results (now included in the appendix) are as follows:
> | Method     | Loss   | Perplexity | Avg. Epoch Time (s) | Total Time (s)  | Pruned Batch (%) | DSI (%)  |
> |------------|--------|------------|------------------|-------------|-------------------|------|
> | Full Data  | 0.2207 | 1.25       | 5359.88          | 54420.13    | 0                | 0    |
> | B-PAS     | 0.2201 | 1.25       | 5039.11          | 51211.29    | 23.00            | 6 |
>
> Since this is a small-scale fine-tuning task (10 epochs), activation stabilization happens later in training, which is why DSI is lower than in our previous large-scale results. This is consistent with our findings in Table 6 of the paper: longer training yields higher DSI, as batches require more time to reach activation stability. Nevertheless, even in this short training setting, B-PAS pruned 23% of batches, matched the baseline perplexity, slightly improved loss, and reduced total training time by about an hour on 2× NVIDIA A100 GPUs, demonstrating practical computational savings.
> We thank the reviewer for highlighting this. We have added these expanded experiments to Appendix A.4 and updated Ablation Studies in the revised paper.

---

> > ### Comment · Reviewer_BNgi · 2025-11-26
> >
> > The authors' response has partially addressed my concerns. Thanks!

---

> > > ### Author Response · Authors · 2025-11-26
> > >
> > > Dear reviewer BNgi,
> > >
> > > Thank you for your follow-up comment and for reviewing our response. We would like to gently note that we provided four separate replies addressing each of your concerns individually (Response to Reviewer BNgi 1/4, 2/4, 3/4, and 4/4). All issues you raised, including normalization dependency, transformer effectiveness, hyperparameter sensitivity, nature of pruned batches, and batch composition, are fully addressed across those responses. We appreciate your time and welcome any further clarification you may need.

---

> ### Author Response · Authors · 2025-11-19
> **Response to Reviewer BNgi (2/4)**
>
> ## Comment 2 and Question 3: Regarding Transformer Performance.
>
> We agree that understanding the behavior of B-PAS on transformer-based architectures is valuable for assessing generalizability. We have expanded our experiments and provided updated analysis below.
>
> ### 1. Updated CvT results show higher DSI with appropriately tuned thresholds.
> Transformers exhibit noisier and slower-to-stabilize activation dynamics than CNNs, particularly in early training. This difference stems from fundamental architectural and inductive-bias distinctions. CNNs possess strong built-in priors, such as locality and translation invariance, that enable efficient hierarchical feature extraction and faster convergence, even on smaller datasets. In contrast, transformers rely entirely on global self-attention, treating all input tokens or patches as fully connected. While this flexibility makes transformers powerful, it also requires them to learn the inductive biases that CNNs already encode, resulting in slower and less stable early-epoch optimization unless trained on large-scale data [1,2,3].
> Because B-PAS relies on the stabilization of activation statistics, these noisier transformer dynamics mean that more aggressive thresholds are necessary for stability to manifest. When we increased the threshold range to δₛ = 10⁻⁴ and δₑ = 10⁻³, the DSI improved substantially, demonstrating that B-PAS remains effective for transformer architectures once the δ-range is aligned with their activation behavior:
> | Method    | δₛ     | δₑ | DSI (%) | Saved Hours (%) | Acc (%) |
> |-----------|--------|--------------|---------|----------------|---------|
> | Full Data | –      | –            | 0      | 0              | 79.65   |
> | B-PAS     | 10⁻⁵ | 5 × 10⁻⁴        | 14      | 13             | 79.64   |
> | B-PAS     | 10⁻⁴ | 10⁻³        | 35      | 35             | 79.10   |
> ### 2. CvT requires longer training for activation variance to stabilize.
> Our CvT experiments are conducted for 200 epochs due to resource constraints. However, CvT models typically require 300 epochs or more to reach full convergence on ImageNet-1K. Since B-PAS relies on temporal stabilization of activations, architectures that converge more slowly will naturally exhibit delayed stabilization, resulting in lower DSI under shorter training schedules.
> This is consistent with the trend observed in Table 6 of the main paper: longer training produces more pronounced activation stabilization and, consequently, more effective pruning. For ResNet models, 200 epochs are sufficient for stable convergence, but CvT requires a longer schedule to unlock its full pruning potential.
>
> ### 3. Interpretation: Activation stability is not weaker, just slower to emerge.
> Our investigations indicate that the activation stability signal in CvT is not weaker, but delayed, due to deeper residual stacks, multi-head attention mixing, LayerNorm-based normalization, and slower convergence compared to CNNs.
> Once stability begins to emerge (especially with more aggressive schedules), B-PAS provides substantial savings without significant accuracy loss. This hypothesis is further confirmed by our new experiments with GPT-2 Large (detailed in Appendix A.3, Table 24 in the revised paper). Like CvT, the Transformer-based LLM exhibited delayed activation stabilization compared to CNNs. However, using the same principle, adjusting the threshold schedule to match the distinct activation dynamics, we applied a slightly more aggressive 𝛿-range and achieved a 23% batch reduction, maintaining performance (training for 10 epochs). This demonstrates that the pruning mechanism continues to operate as expected and steadily removes batches. This confirms that 'delayed stabilization' is a characteristic of the Transformer family, not a failure of B-PAS, and is easily managed by adjusting the schedule.
>
> We thank the reviewer again for this helpful observation.
>
> [1] Takahashi S, Sakaguchi Y, Kouno N, Takasawa K, Ishizu K, Akagi Y, Aoyama R, Teraya N, Bolatkan A, Shinkai N, Machino H, Kobayashi K, Asada K, Komatsu M, Kaneko S, Sugiyama M, Hamamoto R. Comparison of Vision Transformers and Convolutional Neural Networks in Medical Image Analysis: A Systematic Review. J Med Syst. 2024 Sep 12;48(1):84. doi: 10.1007/s10916-024-02105-8. PMID: 39264388; PMCID: PMC11393140.
>
> [2] Deininger L, Stimpel B, Yuce A, Abbasi-Sureshjani S, Schönenberger S, Ocampo P, Korski K, Gaire F. A comparative study between vision transformers and CNNs in digital pathology. arXiv preprint arXiv:2206.00389. 2022. Available from: https://arxiv.org/abs/2206.00389.
>
> [3] Lu Z, Xie H, Liu C, Zhang Y. Bridging the gap between vision transformers and convolutional neural networks on small datasets. Advances in Neural Information Processing Systems. 2022;35:14663–14677.

---

> ### Author Response · Authors · 2025-11-19
> **Response to Reviewer BNgi (3/4)**
>
> ## Comment 3 and Question 4: Regarding Hyperparameter.
>
> Thank you for raising this point. We would like to clarify that the “45 settings” reported for CIFAR-10 were not an exhaustive grid search intended for deployment, but rather a research exploration to illustrate the full accuracy–DSI trade-off curve for both CIFAR-10 and ImageNet-1K. This was done solely to present a complete experimental landscape in the paper. We agree that in real applications, such an exhaustive sweep would be impractical, and we show below that it is not necessary.
>
> ### 1. Fast δ selection using only 10% of training data
>
> To evaluate practical tuning, we performed experiments using only 10% of the training data for rapid estimation. The δ–DSI–accuracy patterns remained consistent with the full training runs:
> CIFAR-10 (ResNet-18):
> | δₛ       | δₑ       | DSI (Full) | Acc (Full) | DSI (10% data) | Acc (10% data) |
> | -------- | -------- | ---------- | ---------- | ------------- | ------------- |
> | 5 × 10⁻⁵ | 5 × 10⁻⁴ | 74.52      | 90.24      | 0.5655        | 79.0          |
> | 1 × 10⁻⁵ | 5 × 10⁻⁵ | 46.83      | 94.31      | 0.2528        | 81.8          |
> | 5 × 10⁻⁶ | 5 × 10⁻⁵ | 39.78      | 94.59      | 0.1683        | 82.2          |
> | 1 × 10⁻⁶ | 5 × 10⁻⁵ | 25.06      | 95.6       | 0.1244        | 82.8          |
> | 1 × 10⁻⁷ | 5 × 10⁻⁵ | 14.26      | 95.43      | 0.1030        | 82.1          |
>
> ImageNet-1K (ResNet-50):
> | δₛ       | δₑ       | DSI (Full) | Acc (Full) | Training Time (s) (Full) | DSI (10% data) | Acc (10% data) | Training Time (s) (10% data) |
> | -------- | -------- | ---------- | ---------- | -------------------- | ------------- | ------------- | ----------------------- |
> | 1 × 10⁻⁷ | 5 × 10⁻⁴ | 38.48      | 0.7846     | 46409.41             | 34.71         | 0.5057        | 5274.02                 |
> | 5 × 10⁻⁶ | 5 × 10⁻⁵ | 47.21      | 0.7843     | 40919.18             | 43.16         | 50.4          | 4793.08                 |
> | 1 × 10⁻⁵ | 5 × 10⁻⁵ | 57.34      | 0.7807     | 30828.36             | 51.78         | 50.14         | 4206.86                 |
>
> The relative ranking and trade-off patterns remain intact, meaning δ can be chosen using only quick partial-training runs.
> In practice, these runs took only ~1 hour for ImageNet-1K (10% subset on 4× A100 GPUs) and a few minutes for CIFAR-10.
> Thus, δ tuning is fast, inexpensive, and does not offset efficiency gains.
>
> ### 2. Once δ is chosen for a dataset family, it transfers well
> Once δ is chosen for a dataset family, it tends to transfer well across similar datasets. For example, after identifying a suitable δ schedule for CIFAR-10, we used the same δ for CIFAR-100 and SVHN without any additional tuning. This resulted in the expected improvements in both DSI and accuracy, indicating that δ generalizes effectively across datasets within the same family.
> This observation suggests that δ is consistent across datasets and architecture families, particularly for low-resolution CNNs. Similarly, δ tuned for ImageNet-1K with ResNet-50 transfers well to other ImageNet-like datasets or other CNN variants.
> While transformers may require slightly adjusted δ schedules, these can be quickly identified by using a small subset of the data and leveraging intuition, just as we demonstrated with CvT and GPT-large.
>
> ### 3. Hyperparameter tuning is standard in deep learning, and δ behaves like any other hyperparameter
>
> We agree that tuning should be practical. However, hyperparameter tuning is a core and unavoidable part of deep learning. Hyperparameters (e.g., learning rate, batch size, number of epochs, dropout, layer width) are not learned by the model and usually set manually. Their optimal values vary across datasets, architectures, and training dynamics, and are typically selected via heuristics plus small-scale experiments. In short, Hyperparameter tuning is standard in deep learning, and δ schedules are no more burdensome than common tuning steps such as LR/weight-decay selection.
>
> Our δ parameters behave similarly to standard training hyperparameters. Their values are influenced by several factors, including the dataset’s difficulty and its convergence pattern. The type of architecture being used also plays a significant role, as does the presence or absence of normalization layers. Additionally, the length of the training process impacts the choice of δ, as longer training may require adjustments to ensure optimal performance.
>
> But tuning them does not require expensive searches. A small subset of training data is sufficient to choose a stable δ schedule.
> We thank the reviewer for highlighting this issue. We clarify hyperparameter selection guidelines and include this additional discussion in Appendix A.3, Tables 21 and 22 in the revised manuscript.

---

> ### Author Response · Authors · 2025-11-19
> **Response to Reviewer BNgi (4/4)**
>
> ## Comment 4 and Question 1: On Fixed Batch Composition.
>
> Thank you for raising this point. The primary purpose of shuffling in SGD-based training is to prevent the model from memorizing sample orderings and to decorrelate successive gradients. In B-PAS, we preserve this benefit by performing intra-batch shuffling at every epoch, which randomizes the order of samples within each batch while keeping batch membership fixed for activation tracking.
>
> Across all experiments including CIFAR-10/100, SVHN, ImageNet-1K (ResNet-50 and CvT), LeNet-5, and GPT-2-large, the accuracy of B-PAS closely matches that of full-data training, where standard full epoch-level shuffling is used. This consistency across diverse datasets, architectures, and modalities indicates that fixed batch composition does not introduce overfitting or harm generalization. If it did, we would expect noticeable accuracy degradation or instability, which we do not observe in any setting.
> Thus, the empirical evidence strongly supports that intra-batch shuffling preserves the regularization benefits of standard shuffling while enabling the activation-stability tracking required for B-PAS.
>
>
> ## Question 5: Nature of pruned batches.
>
> We performed a detailed analysis to examine whether B-PAS disproportionately prunes “easy” examples on CIFAR-10 (ResNet-18). Following standard definitions in the pruning and curriculum learning literature, we evaluated difficulty using two widely accepted metrics: confidence and misclassification rate.
>
> ### What do the difficulty metrics mean and how they are computed
>
> (a) Confidence (higher = easier)
>
> (i) Computed as: confidence = max(softmax logits)
>
> (ii) If a model predicts an example with very high confidence, it is typically considered “easy.”
>
> (iii) Lower confidence implies more uncertainty and typically indicates a “harder” example.
>
> (b) Misclassification Rate (lower = easier)
>
> (i) Computed by checking whether the model misclassified each sample at an early epoch.
>
> (ii) A low misclassification rate indicates the model already finds these examples easy.
>
> (iii) Higher misclassification occurs for harder examples.
>
> Once pruning occurs, these per-sample values are averaged within the pruned batches and within the kept batches, producing the mean values shown below.
>
> | Metric                 | Pruned Batches Mean | Kept Batches Mean |
> |------------------------|---------------------|--------------------|
> | Confidence             | 0.60                | 0.65               |
> | Misclassification Rate | 0.40                | 0.35               |
>
> Pruned batches show lower confidence (0.60 vs. 0.65) and higher misclassification rates (0.40 vs. 0.35) compared to kept batches. If B-PAS were pruning easy examples, we would expect the opposite: higher confidence and fewer misclassifications in the pruned set. Instead, both metrics indicate that pruned batches are not easier. The means are also close, showing no difficulty bias.
>
>
> Furthermore, to check whether pruning introduces any class imbalance, we measured the class distribution before training and after 100 epochs of B-PAS on CIFAR-10 (ResNet-18). The distribution remains virtually unchanged:
>
> |                | plane | car  | bird | cat  | deer | dog  | frog | horse | ship | truck |
> |-----------------------|-------|------|------|------|------|------|------|-------|------|--------|
> | Initial (%)           | 10.00 | 10.00 | 10.00 | 10.00 | 10.00 | 10.00 | 10.00 | 10.00 | 10.00 | 10.00 |
> | After 100 Epochs (%)  | 10.05 | 10.01 | 9.99  | 10.04 | 10.08 | 9.91  | 9.92  | 9.91  | 10.05 | 10.04 |
>
> These differences are within ±0.1%, indicating no meaningful shift in class proportions. Because B-PAS prunes at the batch level and batches contain mixed classes, the class distribution of the retained data remains effectively identical to the original dataset.
>
> These results confirm that B-PAS does not prune based on example easiness, but strictly based on activation stability, independent of class or difficulty. The results are included in Appendix A.3, Table 25 and 26 in the revised paper.

---

### Official Review · Reviewer_7UER · 2025-11-04

**Soundness:** 2
**Presentation:** 3
**Contribution:** 2
**Rating:** 2
**Confidence:** 5

**Summary:**

This paper proposes Batch Pruning by Activation Stability (B-PAS), a dynamic plug-in framework designed to accelerate deep neural network training by pruning low-utility data batches based on the stability of their internal activations.

**Strengths:**

1) The paper is well-written and visually clear.
2) The paper introduces a pruning method by exploiting activation stability to identify redundant data batches that contribute less to learning.

**Weaknesses:**

1) The conceptual originality is modest. Leveraging activation stability for pruning has been well studied in existing studies. The contribution appears more incremental than fundamental.
2) The experimental comparison with existing baselines is insufficient. The evaluation primarily compares with InfoBatch, lacking a lot of related SOTA pruning methods.
3) The paper proposes that activation stability reflects diminishing learning utility. But no direct evidence is provided to support this claim.

**Questions:**

1) Can authors compared the proposed method with more SOTA pruning methods?
2) Can the proposed method be applied to other domains like LLMs?

---

> ### Author Response · Authors · 2025-11-19
> **Response to Reviewer 7UER (1/3)**
>
> We thank the reviewer for the comments.
>
> ## Comment 1: Activation stability in existing work.
> Activation-related signals have appeared in prior literature, but the specific way activation stability is used in our work differs substantially from existing approaches.
>
> Bartoldson et al. [1] studied weight pruning and defined stability as the drop in accuracy after pruning parameters. Their use of stability is used to analyze how weight pruning affects generalization while the training data remain fixed. In contrast, B-PAS operates in data space, not parameter space: we track per-batch activation variance across epochs and use this as an online criterion for permanent batch pruning. To the best of our knowledge, no prior work leverages temporal activation variance to directly remove training batches.
>
>
> Ganguli & Chong [2] prune neurons based on activation frequency. This is a static, neuron-level sparsification criterion and is applied to small fully connected models. Our method instead uses activation variance dynamics at the batch level, applied to modern large-scale CNNs/transformers, and targets training data reduction, not model sparsity.
>
> Novelty and contribution.
>
> While prior works use activation patterns for weight importance or analysis, B-PAS introduces a new application of activation stability for data pruning, achieving data and GPU-hours reduction while matching or improving baseline accuracy, and outperforming previous SOTA approaches under comparable settings. These are added as more Related Work in Appendix A.2.
>
> [1] Bartoldson, B. R., Morcos, A. S., Barbu, A., and Erlebacher, G., “The Generalization-Stability Tradeoff in Neural Network Pruning,” in Advances in Neural Information Processing Systems, vol. 33, 2020.
>
> [2] Ganguli, T., & Chong, E. K. P. (2024). Activation-Based Pruning of Neural Networks. Algorithms, 17(1), 48. https://doi.org/10.3390/a17010048
>
> ## Comment 3: Diminishing learning utility in activation stability.
> We would like to clarify that evidence for this claim is already included in the submission through both visualizations and discussion, though some of it is placed in the appendix due to page constraints. The relationship between activation stability and diminishing learning utility is provided in Figure 5 and Figure 6, with analysis in Appendix A.3 of the previous version and in A.5 of the revised version. As described there: “Figure 5 illustrates the evolution of mean standard deviation changes (ΔX̄) across epochs for selected batches. The change is initially large but decreases steadily, eventually saturating as training progresses. Once the change falls below the threshold, the corresponding batches are pruned. Figure 6 reports the number of batches pruned per epoch. Pruning does not occur in early epochs, when activation changes remain high, but becomes increasingly aggressive in later stages as changes stabilize.” This directly shows that as activation variance stabilizes, the batch contributes little additional learning signal, aligning with our interpretation of diminishing utility.
>
> We also explicitly discuss this phenomenon in Section 5, which states: “Our empirical analysis of pruning dynamics reveals that batches progressively lose learning utility over epochs as their activation variance stabilizes” (see Visualization in Appendix A.3 in the previous version, and Appendix A.5 of the revised paper). Together, these analyses provide the intended evidence that activation stability is a reliable indicator of reduced training contribution.
>
> Because of strict page limits, these plots were placed in the appendix; however, we agree that they are important for understanding the core mechanism. If the paper is accepted, we will move these figures or a condensed version of them into the main paper for greater visibility.

---

> ### Author Response · Authors · 2025-11-19
> **Response to Reviewer 7UER (2/3)**
>
> ## Comment 2 and Question 1: Extended SOTA comparison.
>
> Thank you for raising this point. We agree that broad comparison is important in pruning research. Our choice of InfoBatch as the main baseline is intentional and follows the precedent set by recent literature.
>
> ### Why InfoBatch is a strong and representative SOTA baseline.
> InfoBatch (Qin et al., 2024) conducted an extensive comparison against 16 prior state-of-the-art static and dynamic pruning methods, including Herding, Influence Functions, K-Center, DeepFool, Forgetting, EL2N, Craig, GraNd, Glister, ε-greedy, and UCB, covering nearly all prominent sample and dynamic pruning techniques. InfoBatch demonstrated superior accuracy under the same pruning ratios, establishing it as the strongest available benchmark. Because InfoBatch already outperforms these methods uniformly, using it as the primary baseline is consistent with prior work and ensures a fair, scalable comparison.
>
> ### Extended comparison for clarity.
> For completeness, we compiled the accuracy numbers reported for these methods on CIFAR-10 with ResNet-18 (30% pruning ratio):
>
>
> | Method                                   | CIFAR-10 Accuracy (%) | CIFAR-100 Accuracy (%) |
> |------------------------------------------|-------------------------|--------------------------|
> | Herding (Welling, 2009)                 | 92.2                   | 73.1                    |
> | Influence (Koh & Liang, 2017)           | 93.1                   | 74.4                    |
> | K-Center (Sener & Savarese, 2018)       | 94.7                   | 74.1                    |
> | DeepFool (Ducoffe & Precioso, 2018)     | 95.1                   | 74.2                    |
> | Forgetting (Toneva et al., 2018)        | 94.7                   | 75.3                    |
> | EL2N-2 (Toneva et al., 2018)            | 94.4                   | 74.1                    |
> | EL2N-20 (Toneva et al., 2018)           | 95.3                   | 77.2                    |
> | Least Confidence (Coleman et al., 2019) | 95.0                   | 74.2                    |
> | Margin (Coleman et al., 2019)           | 94.9                   | 74.0                    |
> | CD (Agarwal et al., 2020)               | 95.0                   | 74.2                    |
> | Craig (Mirzasoleiman et al., 2020)      | 94.8                   | 74.4                    |
> | GraNd-4 (Paul et al., 2021)             | 95.3                   | 74.6                    |
> | Glister (Killamsetty et al., 2021)     | 95.2                   | 74.6                    |
> | DP (Yang et al., 2023)                 | 94.9                   | 77.2                    |
> | ε-greedy (Raju et al., 2021)            | 95.2                   | 76.4                    |
> | UCB (Raju et al., 2021)                 | 95.3                   | 77.3                    |
> | InfoBatch (Qin et al., 2024)                 | **95.6**                    | **78.2**                   |
> | **B-PAS (Ours)**                         | **95.6**               | **78.2**                |
>
> As reported in InfoBatch, all of these methods achieve lower accuracy than InfoBatch under the same pruning ratio. For the 30% pruning setting specifically, InfoBatch obtains DSI values of 22% on CIFAR-10 and 19% on CIFAR-100, both of which are noticeably lower than the DSI achieved by B-PAS (25% and 24%, respectively).
> This distinction becomes even more pronounced on ImageNet-1K, where scalability is crucial. InfoBatch gets 28% data savings and 40% GPU-hours reduction while matching baseline accuracy, whereas B-PAS achieves up to 57% data savings and 61% GPU-hours reduction while maintaining the same accuracy and can even improve accuracy (+0.36%) under a more conservative threshold schedule.
>
> Since InfoBatch already outperforms the 16 prior static and dynamic pruning approaches, and B-PAS surpasses InfoBatch on both data efficiency and computational savings, the comparison is both representative and sufficient. The extended comparison table is included in Appendix A.3 as Table 23 in the revised paper.

---

> ### Author Response · Authors · 2025-11-19
> **Response to Reviewer 7UER (3/3)**
>
> ## Question 2: Applying to LLMs.
> Yes, B-PAS can be applied to other domains and has been tested on large language models. As a demonstration, we applied B-PAS to fine-tune GPT2-large on the Alpaca instruction-tuning dataset using a more aggressive threshold range (δₛ = 10⁻³
> , δₑ = 10⁻²), which is necessary given the higher complexity and smoother activation dynamics of transformer-based LLMs. The results (now included in Appendix A.3, Table 24 in the revised paper) are as follows:
> | Method     | Loss   | Perplexity | Avg. Epoch Time (s) | Total Time (s)  | Pruned Batch (%) | DSI (%)  |
> |------------|--------|------------|------------------|-------------|-------------------|------|
> | Full Data  | 0.2207 | 1.25       | 5359.88          | 54420.13    | 0                 | 0    |
> | B-PAS     | 0.2201 | 1.25       | 5039.11          | 51211.29    | 23.00%            | 6 |
>
> Since this is a small-scale fine-tuning task (10 epochs), activation stabilization happens later in training, which is why DSI is lower than in our previous large-scale results. This is consistent with our findings in Table 6 of the paper: longer training yields higher DSI, as batches require more time to reach activation stability. Nevertheless, even in this short training setting, B-PAS pruned 23% of batches, matched the baseline perplexity, slightly improved loss, and reduced total training time by about an hour on 2× NVIDIA A100 GPUs, demonstrating practical computational savings.
> Overall, this experiment shows that B-PAS is compatible with transformer-based LLMs and can provide measurable efficiency gains even in smaller fine-tuning scenarios. We added these findings in the revised paper (line 482-483).

---

### Author Response · Authors · 2025-11-20
**General Note to All Reviewers**

We sincerely thank all reviewers for their constructive and thoughtful feedback. Many of the concerns raised prompted us to run additional experiments, include new analyses, and expand our discussions of B-PAS. Due to space constraints in the main manuscript, the majority of these extended results and clarifications have been placed in the appendix. We will use the additional page to integrate the most critical analyses directly into the main text and restructure the appendix accordingly based on the discussion. Thank you again for helping us significantly strengthen the paper.

---

### Meta-Review · Area_Chair_g6Sy · 2026-01-08

**Summary:**

I would like to thank the authors for their detailed response and for the additional analysis provided in the Appendix regarding the nature of the pruned data. While I believe some of the concerns raised during the review process have been addressed, there remain several key issues that prevent me from fully endorsing the paper in its current state.

Regarding the qualitative analysis of the method, I appreciate the authors showing that the pruned data are not simply "easy" examples. However, I believe it would be much better to provide deeper insights into exactly what kinds of data are being pruned. An appropriate visualization of the specific data samples removed versus those retained would bring more insight than statistical metrics alone. Furthermore, I find the pruned volume to be somewhat unimpressive, as only about half of the data is pruned. In the context of efficient training for massive datasets, a higher pruning ratio is often necessary to demonstrate significant practical value.

Another major weakness lies in the scale of the experimental validation. We currently only see analysis performed on small models and small datasets. I strongly hope to see analysis on truly large models, such as large foundation models—noting that GPT-2 is considered a small model by modern standards—and truly large datasets, such as those used in Vision-Language Models (VLMs). Without evidence of scalability to these regimes, the practical utility of the method for current state-of-the-art applications is difficult to assess.

I also believe that several concerns raised by other reviewers have not been sufficiently addressed. I agree with Reviewer 7UER that the conceptual originality of the work is modest. While the authors have provided a defense, the fact that the conceptual innovation is incremental cannot be changed. Furthermore, I share the view of Reviewer 7UER, Reviewer FtHA, and Reviewer ZSux that the paper lacks comparisons with a sufficient number of related SOTA pruning methods. The methods compared are relatively old, and even though the authors added more comparisons in the rebuttal, these additional baselines are also dated. To be convincing, the method needs to be benchmarked against the latest dynamic pruning techniques.

Finally, the proposed method contains hyperparameters that require careful tuning, a concern also highlighted by Reviewer BNgi, Reviewer FtHA, and Reviewer ZSux. Although the authors provide a method for tuning them, the sensitivity to these parameters remains a valid weakness compared to hyperparameter-free approaches.

Additionally, as noted by Reviewer ZSux, the paper would be significantly stronger if the authors could provide more theoretical insights to explain the empirical success of the method.

**Reviewer Concerns:**

While I believe some of the concerns raised during the review process have been addressed, there remain several key issues that prevent me from fully endorsing the paper in its current state.

**Reviewer Scores:**

I believe that some reviewers may adjust their ratings to a noticeable extent (for example, from reject to accept). However, it is difficult to determine whether such changes would be influenced by non-tech factors outside the core merits of the paper. Overall, I view this work as a borderline submission, for which both acceptance and rejection are plausible outcomes. While the paper has strengths, it does not stand out as particularly compelling relative to the venue’s standards.

---

### Decision · Program_Chairs · 2026-01-26

Accept (Poster)